# High resolution stochastic downscaling method for ocean forecasting models and its application to the Red Sea dynamics

Georgy I. Shapiro[1], Jose M. Gonzalez-Ondina[2], Vladimir N. Belokopytov[3]

[1]School of Biological and Marine Sciences, University of Plymouth, Plymouth, PL4 8AA, UK
5   [2]University of Plymouth Enterprise LTD, Plymouth, PL4 8AA, UK
[3]Marine Hydrophysical Institute, Russian Academy of Sciences, Sevastopol, 299011, Russia

*Correspondence to*: Georgy I. Shapiro (gshapiro@plymouth.ac.uk)

**Abstract**. High-resolution modelling of a large ocean domain requires significant computational resources. The main purpose of this study is to develop an efficient tool for downscaling the lower resolution data such as available from Copernicus Marine Environment Monitoring Service (CMEMS). Common methods of downscaling CMEMS ocean models utilize their lower resolution output as boundary conditions for local, higher resolution hydrodynamic ocean models. Such methods reveal greater details of spatial distribution of ocean variables; however, they increase the cost of computations, and often reduce the model skill due to the so called 'double penalty' effect. This effect is a common problem for many high-resolution models where predicted features are displaced in space or time. This paper presents a Stochastic-Deterministic Downscaling (SDD) method, which is an efficient tool for downscaling of ocean models based on the combination of deterministic and stochastic approaches. The ability of the SDD method is first demonstrated in an idealised case when the true solution is known a priori. Then the method is applied to create an operational Stochastic Model of the Red Sea (SMORS) with the parent model being the Mercator Global Ocean Analysis and Forecast System at 1/12th degree resolution. The stochastic component of the model is data-driven rather than equation-driven and it is applied to the areas smaller than the Rossby radius, within which distributions of ocean variables are more coherent than over a larger distance. The method, based on objective analysis, is similar to what is used for data assimilation in ocean models, and stems from the philosophy of 2D turbulence. The SMORS model produces finer resolution (1/24[th] degree latitude mesh) oceanographic data using the output from a coarser resolution (1/12[th] degree mesh) parent model available from CMEMS. The values on the fine-resolution mesh are computed under condition of minimisation of the cost function which represents the error between the model and true solution. The SMORS model has been validated against Sea Surface Temperature and ARGO floats observations. Comparisons show that the model and observations are in good agreement and SMORS is not subject to the 'double penalty' effect. SMORS is very fast to run on a typical desktop PC and can be relocated to another area of the ocean.

## 1 Introduction

The main aim of this paper is to present an alternative, computationally efficient method of downscaling of ocean models, i.e. create finer resolution outputs using a stochastic method while the coarser resolution fields are obtained by a traditional deterministic numerical ocean modelling. In order to reflect the dual nature of the algorithm, the term 'Stochastic-deterministic' is used. The suggested method may do best in going from eddy-permitting resolution where the desired features are "already" there embryonically and guided by assimilation (e.g. as in CMEMS) to somewhat finer resolution so that the embryonic features can be properly represented. As usual, the method has its limitations which are discussed later. A deterministic approach in ocean modelling based on solving differential equations is capable of producing high quality forecasts/hindcasts, both for research and operational needs and is currently a mainstream in numerical modelling of the ocean. Ocean models have matured through multiple improvements including better numerical schemes, spatial discretization, parameterizations, and data

assimilation. Modern ocean models do not solve the full Navier-Stokes or Reynolds equations, instead they tend to make the traditional and hydrostatic Boussinesq approximations and various parameterisations of unresolved processes (Miller, 2007; Fox-Kemper et al., 2019; Lindsay, 2017; Ezer and Mellor, 2004; Bruciaferri et al, 2019).

However, the enhancement of model resolution using such approach involves a significant increase in the computational cost. For example, doubling the horizontal resolution in both directions requires approximately ten times more calculations, taking into account the necessity of reducing the time step and increasing the overhead due to data exchange between the nodes of a High Performance Computer. There is also an increased conceptual difficulty to resolve deterministically very small-scale processes due to the turbulent and chaotic nature of motion at a small scale.

In contrast to early ocean models which were applied to highly idealized cases and did not require any observational data, e.g.
(Bryan, 1963), modern models use real-world data in addition to the universal laws of physics. The data are used for model initialisation, tuning the numerical parameters such as diffusion/viscosity coefficients, validation and data assimilation. Data assimilation improves the description of ocean state used as the initial condition for the forecasting step. There are many different forms of data assimilation including Optimal Interpolation (OI), Kalman filtering and variational methods, see e.g. (Lorenc, 1986) and references therein. One of the most efficient methods is optimal interpolation (OI) (Gandin 1959; Gandin
1965; Fletcher, 2017) which uses statistical properties of real-world data rather than equations of motion or prescribed spatial dependences].

The term 'optimal interpolation' may be confusing as it is of a very different nature than the usual deterministic interpolation methods (linear, polynomial, spline, inverse distance etc.) where the weighting coefficients are determined by the location of points, not by the data themselves. In contrast, the OI method calculates the weights based on statistical properties of the data
and could be called 'objective analysis'. However, the term 'objective analysis' has already been occupied in the original publication by Cressman (1959) for his deterministic interpolation method. Therefore this paper follows the terminology from original literature and uses the term 'optimal interpolation' even though it is not strictly interpolation, but a minimum variance estimator that is algorithmically similar to Kalman filtering.

The philosophy of combining deterministic and stochastic (random) behaviour of fluids has a long history. For example the
Reynolds equations and their modern versions are used in ocean modelling, based on simple decomposition of an actual instantaneous quantity into time-averaged and fluctuating quantities and taking the averages of non-linear terms, see e.g. (Tennekes and Lumley, 1992).  More advanced methods of describing the chaotic movements at smaller scale have been developed in the statistical theory of turbulence, see e.g. (Kolmogorov, 1941; Monin and Yaglom, 1971; Frisch, 1995).  The OI method further extends ideas originated in the theory of statistical turbulence and was the method of choice for operational
numerical weather prediction centres in the 1980s and early 1990s. As shown by Lorenc (1986), more modern variational methods are closely linked to the original OI and they can be described using a common Bayesian analysis framework.

The basis of OI is the minimisation of a cost function which represents a measure of the difference between the estimated and true values. The OI considers the data fields as realisations of random processes and it studies the statistical links represented by either structure functions or covariances between data points in a way similar to the theory of fully developed turbulence

(Gandin and Kagan, 1976). An important feature of the method is that, in order to calculate the interpolating coefficients, it only requires the knowledge of statistical moments of the second order. It does not use any *a priori* hypothesis about the dependence of the weights on the distance from the interpolation points as it is used in alternative methods of objective analysis (Cressman 1959, Vasquez, 2003). In those alternative methods the weighting coefficients are calculated as a prescribed analytical function of distance, and hence do not require the knowledge of the statistical properties of the actual field of interest.

In this paper we have tested a hypothesis that a similar technique, hereafter called Stochastic-Deterministic Downscaling, or SDD, based on the statistical properties of ocean parameters such as temperature, salinity and velocity, can be used to achieve a finer resolution in ocean modelling by downscaling the results of a parent deterministic model. Basically, the data are treated as having two components, a low resolution, slowly varying component which is computed using deterministic equations, and a high resolution fast varying component where the data are treated as random processes. As in the theory of turbulence, the

statistical properties of the smaller scale processes are often much more stable than the data themselves, see e.g. (Monin and Yaglom,1971; Tennekes and Lumley, 1992).

The assimilation of observational data is widely used in operational ocean modelling, see e.g. (Dobricic et al., 2007; Dobricic and Pinardi, 2008; Korotaev et al., 2011; Mirouze et al., 2016). However, the application of a similar approach for fine-resolution model downscaling should be considered as experimental at this stage. The SDD method, in common with other

data assimilation techniques, can be used both in the attached and detached modes. In the attached mode the downscaling is carried out on the same computer which solves the equations of ocean dynamics at the same time as the forecast advances. Programmatically, in the attached mode the SDD is contained within the same executable module as all other elements of the model and is applied regularly as the model advances in time. On the other hand, in the detached mode, the SDD is applied after the forecast has been completed by the parent model. This mode was used in SMORS. In this case the SDD (or any data

assimilation) can be considered as post-processing. The treatment of data assimilation as post-processing can be found in (Delle Monache et al., 2011; DazhiYang, 2019) and references therein. Due to its experimental nature, the SDD method is first tested and assessed by application to an idealised case of a region filled with multiple mesoscale eddies where the true solution is known.

While the proposed SDD method has a generic nature, the focus of this paper is on its application to the Red Sea. We use the

Red Sea as a 'difficult' use case for the SDD method as the sea has complicated coastline, multiple islands and a complex structure of its mesoscale circulation, see e.g. . Zhan  et al, 2016; Hoteit et al. 2021 and references therein.  The main section of the paper describes the development and properties of operational eddy-resolving Stochastic Model for the Red Sea (SMORS) at 1/24[th] degree resolution based on a parent eddy-permitting model at 1/12[th] degree resolution, which outputs are accessible via Copernicus Marine Environment Monitoring Service (CMEMS, 2020).

The paper is organised as follows. Section 2 describes materials and methods, including a detailed description of the algorithm used in SDD, application of the method for an idealised case, the treatment of noisy data, and a description of the operational Red Sea model (SMORS). Section 3 presents the results of SMORS model validation, analyses of eddy and mean kinetic

energy as well as analyses of vorticity and enstrophy produced by the parent and SDD models. Section 4 present the discussion of the results and Section 5 concludes the paper.

## 2 Materials and methods

### 2.1 The algorithm

The Stochastic-Deterministic Downscaling (SDD) uses the methodology developed for the original version of the Optimal Interpolation technique (Gandin 1959; Gandin, 1963; Gandin 1965; Gandin and Kagan, 1976; Barth et al., 2008). The philosophy behind this technique is similar to what is used in assimilation of observational data to improve the quality of numerical models. The main differences are that instead of observational data, the SDD assimilates the data from a medium-resolution model, and the effect is the enhancement of model resolution rather than improvement of model skill. The SDD method considers all oceanographic fields as consisted of two components: (i) a relatively slowly varying part which can be described using a dynamic method (i.e.by solving deterministic equations), and (ii) a stochastic, turbulent part which can be described via its statistical properties. Then the statistical properties are linked to the properties of slowly varying field similar to how a turbulent viscosity coefficient is estimated in ocean modelling via the knowledge of deterministically assessed larger scale flows, see e.g. (Smagorinsky, 1963).

We treat the data from the parent model as 'observations' and assimilate these onto a fine-resolution mesh of SMORS. Generally speaking, the OI method requires, among other parameters, the knowledge of the RMS error of 'observations' at each location to calculate the interpolating weights. As the errors of the parent models at each grid point are often not known, we assume that the medium resolution forecast provides the values $f_1 = f(\boldsymbol{r}_1), \dots, f_n = f(\boldsymbol{r}_n)$ for a certain oceanographic parameter $f$ at all points $\boldsymbol{r}_1, \dots, \boldsymbol{r}_n$ on the parent mesh with perfect accuracy (later, in sub-section 2.3 we shall see that this requirement can be relaxed). We are interested in finding the value of the parameter $f$ at another location $f_0 = f(\boldsymbol{r}_0)$ where $\boldsymbol{r}_0$ is any point on a fine resolution mesh. The SDD method is applied to the deviations $f_i' = f'(\boldsymbol{r}_i) = f(\boldsymbol{r}_i) - \langle f(\boldsymbol{r}_i) \rangle$ of the parameter from its statistical mean, or 'norm', designated here as $\langle f \rangle$, rather than to the parameter $f$ itself, in line with the approach used in (Gandin, 1965). We further assume that the field of deviations $f'$ is statistically homogenous and isotropic. This assumption has been shown to be better applicable to the deviations than to the meteorological and oceanographic parameters themselves (Gandin and Kagan, 1976; Fletcher, 2017; Barth et al, 2008). Bretherton et al (1976) have also recommended that for oceanographic applications an estimated mean should be subtracted from each observation at the outset, and added back to the estimate of interpolated values. Climatic studies have also shown that fluctuations (aka anomalies) have better statistical properties than the data itself, and hence it is the statistics of fluctuations rather than full data that are usually used on oceanographic research , see e.g. (Boyer et al 2005)

The calculation of statistically mean value requires averaging over a statistical ensemble, which, as usual, was not available. The estimate of statistical mean of a parameter $\langle f \rangle$ was calculated by computing the spatial average inside the Red Sea of the

values of the parameter in the daily analysis data corresponding to one year (2016). Deviations from climatology would be a satisfactory alternative as well. These daily spatial averages were averaged in time to obtain monthly averages. This means that $\langle f \rangle$ is independent of the location but has a dependency on time since each month has a different norm.

According to (Gandin, 1965), an approximate estimate $\widetilde{f'}_0$ of the true deviation $f'_0 = f'(\mathrm{r}_0)$ at a location $\boldsymbol{r}_0$ can be found as a linear combination of deviations at other points as:

$$\widetilde{f'_0} = \sum_{i=1}^n p_i f'_i, \tag{1}$$

where $p_i$ are the weighting factors that must be determined. This is done by minimising the variance of the difference between the true and estimated values of deviations, also known as a cost function:

$$E = \left\langle \left(f'_0 - \widetilde{f'_0}\right)^2 \right\rangle = \left\langle \left(f'_0 - \sum_{i=1}^n p_i f'_i\right)^2 \right\rangle. \tag{2}$$

The cost function given by Eq. (2) can be rewritten in terms of the autocorrelation matrix

$$R_{ij} = \frac{\langle f'_i f'_j \rangle}{\langle (f'_0)^2 \rangle}, \tag{3}$$

also known as a background error correlation matrix as follows:

$$E = \langle (f'_0)^2 \rangle (1 - 2\boldsymbol{R}_0^T \boldsymbol{p} + \boldsymbol{p}^T R \boldsymbol{p}), \tag{4}$$

where $\boldsymbol{p}$ is the column vector composed by the unknown weighting coefficients $p_i$, $i = 1 \dots n$ and $\boldsymbol{R}_0$ is the column vector of correlations $R_{0i}$ given by Eq. (3). The optimal values of weights $p_i$ which minimise the cost function $E$ given by Eq. (4) can be found by taking partial derivatives of $E$ with respect to all the $p_i$ and equalling them to zero, resulting in the following system of linear equations:

$$R\boldsymbol{p} = \boldsymbol{R}_0. \tag{5}$$

These equations can be solved for the weights $p_i$ if we know the background correlation matrix $R$. Background correlation describes the statistical structure of deviations $f'$ in space and can be found as described below.

Following Gandin (1963), only those correlations which relate to the data located at the same depth level are taken into account, and the distribution of deviations $f'$ is assumed to be statistically uniform and isotropic locally ( i.e. within the search radius, see Eq. (8) below) in the horizontal plane. Therefore, the autocorrelation $R$ matrix can be represented in the form

$$R_{ij} = C\big(\|\boldsymbol{r}_i - \boldsymbol{r}_j\|, z\big), \tag{6}$$

where $\boldsymbol{r}_i$, $\boldsymbol{r}_j$ are horizontal coordinates of the parent grid points, $\|\boldsymbol{r}_i - \boldsymbol{r}_j\|$ is the distance between points $\boldsymbol{r}_i$ and $\boldsymbol{r}_j$ independently of the direction, and $z$ is a vertical coordinate (depth). For 3-dimensional fields, Fu et al. (2004) suggested to approximate the correlation function defined by Eq. (6) using a Gaussian formula which can be written in horizontally isotropic case as follows:

$$R_{ij}\big(\|\boldsymbol{r}_i - \boldsymbol{r}_j\|, z\big) = \exp\left(-\frac{(r_i - r_j)^2}{L(z)^2}\right), \tag{7}$$

where $L(z)$ is the e-folding correlation radius, representing the scale which reflects the extent of spatial correlation, and z is the depth level where correlation is calculated. The use of Gaussian function for the autocorrelation and associated difficulties

have been discussed by R. Daley (Atmospheric Data Analysis 1991). The downscaling process described by Equations (1) —

(7) is repeated for every ocean parameter on every grid point of the fine-resolution mesh, to provide a fine-resolution output

for deviations $f'_i$. The fine-resolution output of the actual values is calculated by adding the deviations to the 'norms'.

To reduce the computational cost while solving multiple systems of equations (5), only those nodes which are relatively close

to the point of interpolation $r_0$ are taken into account, so that the corresponding matrix elements are larger than a certain

threshold. We use a correlation threshold $R_{cut}$ suggested by (Grigoriev et al, 1996) for using the optimal interpolation technique

in the analysis of ocean observations in the Black Sea. Our tests confirmed that this method provides accurate results in the

downscaling of model outputs while avoiding numerical and computational problems. In order to further optimise the

computational algorithm, the correlation threshold $R_{cut}$ was converted into a maximum distance $r_{max}$, which is computed just

once for each depth:

$$r_{max} = L(z)\sqrt{-\ln(R_{cut})} \qquad\qquad\qquad (8)$$

For computation of the correlation matrix $R_{ij}$ using the expression in Eq. (5), it is only necessary to include the nodes in the

medium mesh located at a distance smaller than $r_{max}$ to the fine-resolution node being computed. It is worth noting that the

SDD method honours the data on the coarse grid, i.e. it reproduces the coarse field data exactly (to within truncation errors) at

those fine grid nodes which coincide with the parent grid points. If the fine grid contains the coarse model gridpoints then the

values at this points are exactly the same (to within truncation errors) as in the parent model. Therefore, the spatial structure is

anchored onto the coarse grid and no additional double penalty effect compared to the parent model is generated.

## 2.2 Idealised case

The SDD technique can be illustrated using an idealised case. Let us consider a rectangular domain, which is significantly

larger than a typical size of a mesoscale eddy. In this numerical example we use an area of 1000 km × 1000 km. The parent

model is assumed to produce no errors, with its only limitation being an insufficient resolution. Let the parent grid to have a

spatial resolution of $\Delta x_p = \Delta y_p = 10$ km, and the true 2D field of variable $F$ consist of a number of anisotropic vortices which

are modelled by the formula:

$$F(x,y) = \sin\left(\frac{x}{a}\right)\sin\left(\frac{y}{b}\right), \qquad\qquad\qquad (9)$$

as shown in Fig.1. The statistical norm of $F$ is zero and hence the equations (1)-(7) can be applied to the parameter $F$ itself.

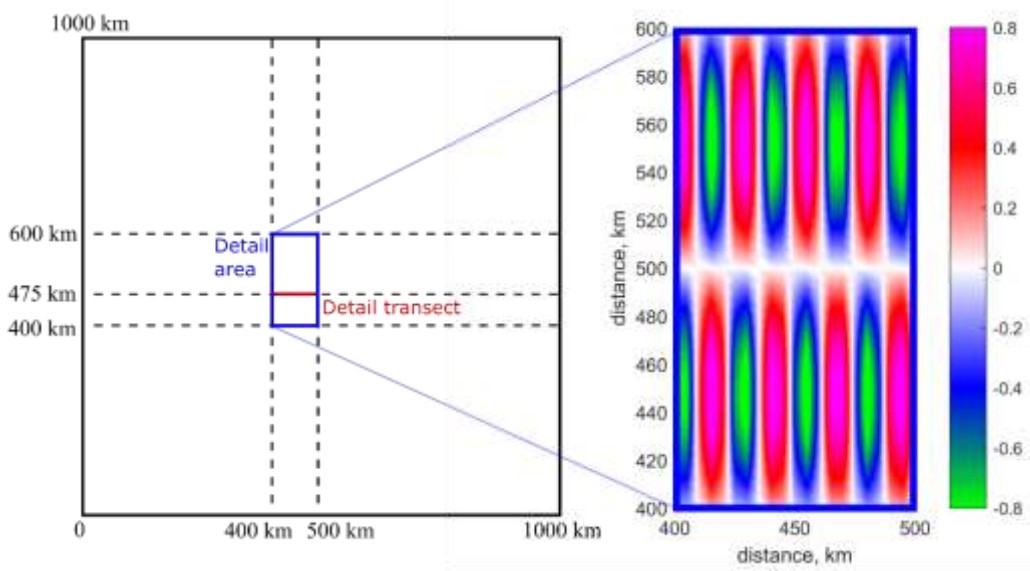


**Figure 1: Model domain with a zoomed-in sub-map of idealised spatial distribution of parameter $F$ according to Eq. (9) with $a = 4.1$ km, which corresponds to the eddy size of 13 km in $x$-direction, and $b = 33.3$ km, which corresponds to the eddy size of 105 km in $y$-direction.**

For this exercise we selected parameters $a$ and $b$ in Eq. (9) so that the parent model can only be considered as eddy-permitting but not eddy resolving, and hence a significant distortion of the true field is expected. The fine-resolution mesh for the SDD model has spatial resolution in each direction twice as high as the parent model, namely $\Delta x_d = \Delta y_d = 5$ km. The correlation matrix is calculated using Eq. (7) with $L = 24$ km for each grid node on the fine mesh. As with many data assimilation methods, e.g. Hollingsworth and Lönnberg (1986), this approach does not require the knowledge of the true solution. The

value of $L$ was obtained using a trial and error method within a range used in OI of observational data (Belokopytov, 2018). Then the linear algebraic system of equations (5) is solved for each fine mesh node, and the final stochastic downscaling is carried out using Eq. (1). In this simple example, the correlation matrices are relatively well-conditioned, with a condition number of the order of $CN = 10^4 — 10^5$ (see section 2.3 for a more detailed discussion on condition numbers.)

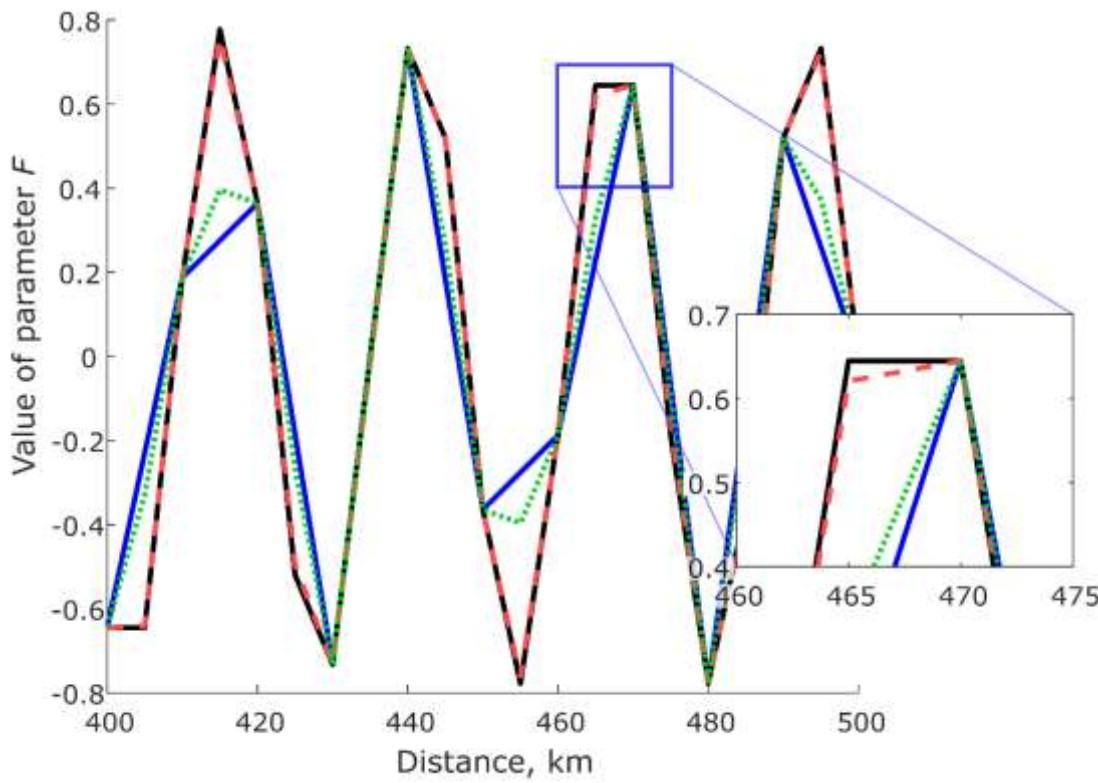

**Figure 2: A zonal transect (see 'detail transect' location in Fig. 1) showing the value of parameter _F_ produced by three models: SDD model (dashed red line) and the coarse model bi-linearly interpolated (blue line) and bi-cubically interpolated (dotted green line) in comparison to the true solution (dashed black line) on the fine grid. Since SDD produces results almost identical to the true solution an inset with a zoomed region has been included to make the differences more clear.**

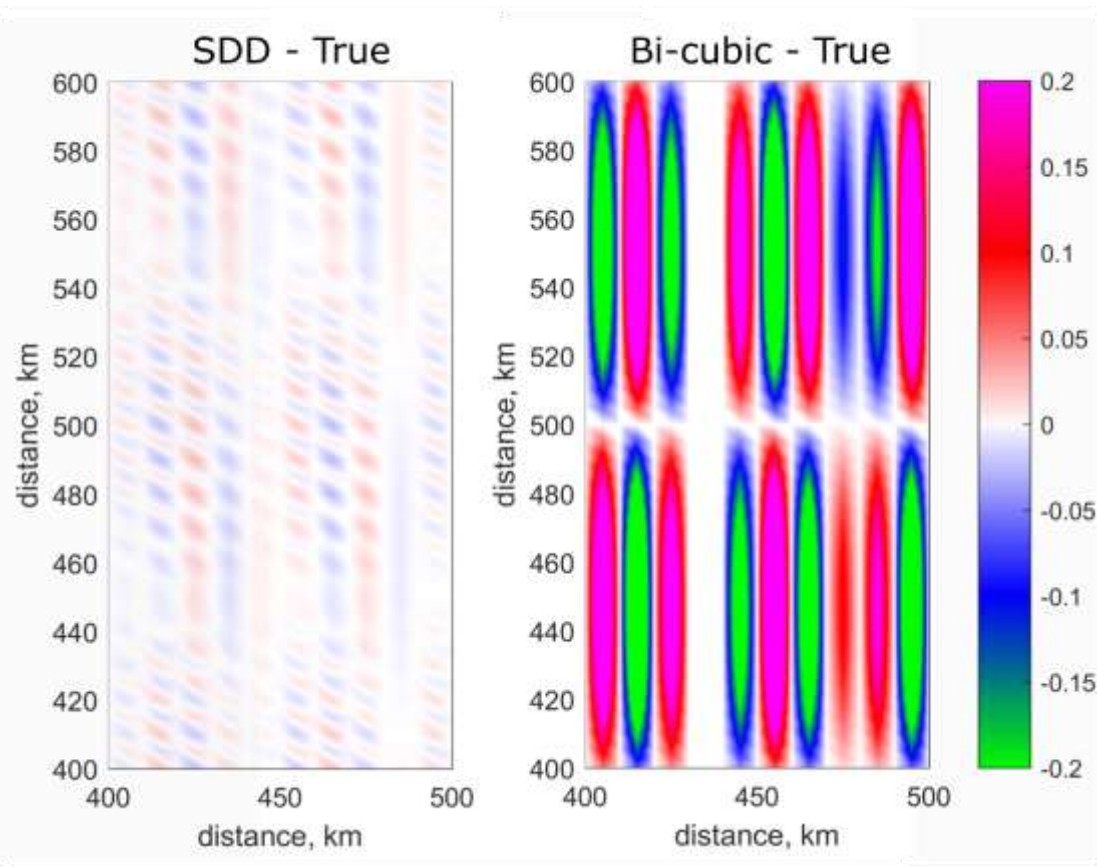


**Figure 3: Comparison of skill between SDD and bi-cubic interpolation: Left: map of differences of parameter _F_ values between SDD model and the true solution. Right: map of differences of parameter _F_ values between bi-cubic interpolation and the true solution. The colour bar limits are chosen to be between $-0.2$ and $0.2$ so the differences for SDD are visible, however, the maximum**

**differences for the bi-cubic model are in fact between $-0.4$ and $0.4$.**

For any point on the fine mesh, the stochastic downscaling uses statistical properties of the data within the surrounding area of influence with size defined by Eq. (8). In this idealised example the surrounding area contains up to 89 points. One could consider an alternative method of enhancing the resolution of the model output by interpolation of the coarse grid using a linear

or polynomial interpolation, which only uses information from a small number of surrounding grid nodes, in a way suggested by (Gilchrist and Cressman, 1954). Another simple alternative would be the use of a prescribed analytical formula for weighting coefficients in Eq. (1) as a function of distance, a method which was widely used in early versions of objective analysis of meteorological fields (Cressman, 1959). However, it was shown, see e.g. (Gandin and Kagan, 1976) that the downscaling method based on Equations (1)—(5) minimises the error between the estimated and the true values of the

parameter and hence better recovers the values in-between the nodes of the coarser eddy-permitting model than the polynomial or similar interpolation methods.

This example gives a quantitative estimate of how much improvement can be achieved by using the SDD method instead of interpolation based on analytical formula. Fig. 2 shows the results produced by the SDD model in comparison with the true solution and two polynomial interpolating models (bi-linear and bi-cubic) along a zonal transect located as shown in Fig. 1. The maps of differences between the true solution, SDD and the bi-cubic model are shown in Fig. 3. All data are sampled on the fine-resolution grid. The SDD model is able to (i) recover the extremes missed in the parent, linear interpolating and bi-cubic interpolating models and (ii) generate a solution that is much closer to the exact one. The root-mean-square error produced by the SDD model is only 0.005 while the error produced by the bi-cubic interpolating model is approximately 35 times higher at 0.177. The SDD method is computationally efficient; it takes only a few seconds to run the fine-resolution model on a small laptop for a 1000 km × 1000 km domain in this idealised setting.

The maximum enhancement produced by SDD compared to simple interpolation is expected when the parent model barely resolves the field. If the ocean feature is well resolved by the parent model, there is no need for further refinement. For example, if the zonal size of the eddy is increased to 40 km instead of 13 km, it is reasonably well resolved by the parent model with $\Delta x = 10$ km. We calculated RMSE using the same value of $L = 24$ km and the results for SDD at $8.1 \times 10^{-3}$ % and bi-cubical interpolation at 0.3% are very small, so downscaling is not actually required, On the other hand, if the parent model misses the features completely, e.g. is not eddy permitting, then the SDD method does not have enough information to re-create the smaller scale features.

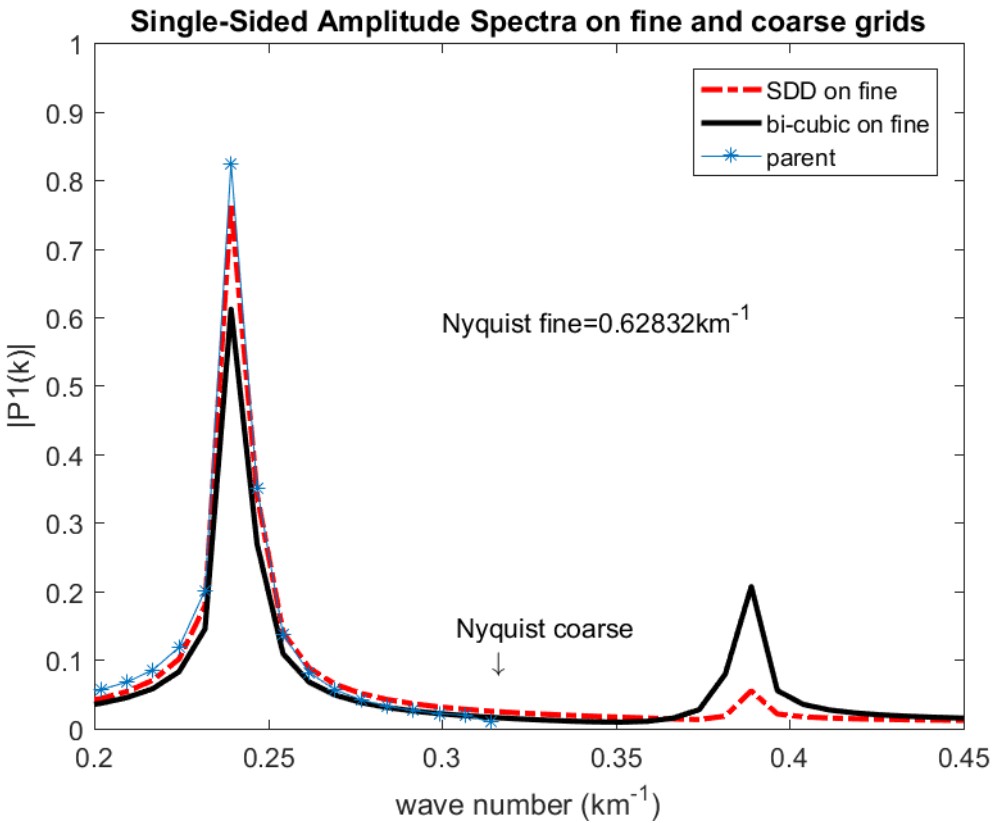

**Fig 4. Amplitude spectra of parameter F on a zonal transect at $y = 40$ km (see map in Fig. 1) based on data from the parent model on the coarse grid (blue-star line), SDD method (red) and bi-cubic interpolation (black). For clarity only the central part of the spectrum showing the peaks is presented.**

The spectral analysis was carried out for the data produced on an 840 km long zonal transect. The Fourier spectrum of the field produced by SDD is close to the spectrum of the true field on the coarse grid (i.e. parent model) up to the Nyquist wavelength of the coarse grid as can be seen in Fig. 4. This figure shows the spectra of the (i) true field on coarse grid, (ii) downscaled with SDD on fine grid, (iii) bi-cubic interpolated onto the same fine grid. The main peak on SDD spectrum is closer to the true peak than that produced by bi-cubic interpolation. In the spectral region between the Nyquist wavenumbers of the coarse and fine grids, there is a parasitic peak that is an artefact caused by distortion of the fields by bi-cubic interpolation as well as downscaled by SDD. However, this artefact is much smaller in the case of SDD which demonstrates its better skill of recovering the true field.

The idealised case where we know the true field gives us some confidence that the additional powers at

high wavenumbers in the real world situation are mainly a representation of the true field not artefacts.

**2.3 Effect of noise in the input data**

Obviously, the data provided by the parent (coarse) model is not precisely correct, it contains errors originated from uncertainties in the input data and errors from the model itself. An ocean model is likely to be less reliable at the grid-scale.

Here we investigate how the noise present in the parent model propagates into the downscaled fine-resolution data by 3 different downscaling methods: (i) SDD, (ii) bi-linear and (iii) bi-cubic interpolation. We use the same idealised field F given by Eq. (9) but with added random uncorrelated Gaussian noise N at each grid point of the coarse grid

$$F_N = F + N$$

We used 4 amplitudes of noise N: 1%, 5% 10%, and 20% of the amplitude of the true field. The magnitude of noise on the fine grid is quantified as the root-mean-square error (RMSE) between the downscaled and exact fields for the 3 downscaled methods. The data show that the SDD method does not increase the noise present in the parent model, whilst both bi-linear and bi-cubic interpolation significantly increase the noise on the fine grid by introducing their own errors in the downscaling process. For example, at a noise level of 5%, the SDD method produces RMSE of 4.8%, bi-cubic produces 18% and bi-linear

gives 24%.

| Noise amplitude in coarse mesh (%) | RMSE after applying SDD (%) | RMSE after applying Bi-cubic (%) | RMSE after applying Bi-linear (%) |
|---|---|---|---|
| 0 | 0.5 | 18 | 23 |
| 1 | 1.1 | 18 | 23 |
| 5 | 4.8 | 18 | 24 |
| 10 | 9.6 | 19 | 25 |
| 20 | 19 | 24 | 28 |

Table 1. RMS errors between the true signal on the fine mesh and the ones obtained by downscaling from the coarse mesh

using different methods. The RMSE is computed in the detailed area shown in Fig. 1.

High resolution reveals more intricate granularity and provides important information of smaller-scale processes, in particularly those dependent of the gradients of the simulated variables. It is known that gradients of noisy data can have greater errors than the variables themselves, see e.g. (Brekelmans et al, 2003). Hence, the finer resolution models should ideally

have better absolute accuracy than the coarser resolution models for the study of such properties as flow vorticity or geopotential gradients related to geostrophic currents. The idealised experiments shown above demonstrate that SDD model not only reveals more small-scale features, but it also improves the accuracy of simulation meaning that the SDD model has the ability to forecast greater granularity, variation, and extremes as compared with commonly used interpolation schemes , e.g. bi-linear or bi-cubic.


## 2.4 Stochastic Model of the Red Sea

In this section the SDD method is applied to create a fine resolution, eddy-resolving model of the Red Sea (SMORS) based on the medium-resolution, eddy-permitting parent model. The parent model used in the study is PSY4V3R1, which is part of the Mercator Global Ocean Analysis and Forecast System based on NEMO v 3.1. The parent model assimilates observational data

and has a medium $1/12^{th}$ degree resolution with 50 depth levels (CMEMS, 2020). The outputs from this model are freely available as Copernicus Marine Environment Monitoring Service product GLOBAL_ANALYSIS_FORECAST_PHY_001_024 (hereafter called PHY_001_024). This product contains daily 10-day forecasts of U- and V-components of current velocities, Temperature and Salinity in 3D and hourly outputs of temperature and currents at the surface in 2D. In addition to the currents produced by PSY4V3R1, the surface hourly currents include tidal

streams and Stokes drifts. The output data are interpolated from the native staggered Arakawa C-grid onto an A-grid. SMORS uses finer resolution bathymetry obtained from the 30 arc-second grid (GEBCO, 2014). The coastline and the land masks at each depth level are obtained from the bathymetry data.

The SMORS downscaling model has a $1/24^{th}$ degree horizontal resolution. We have developed two versions of SMORS: (i) SMORS-3D uses 3D daily outputs from PHY_001_024 as its input, and (ii) SMORS-2D uses surface data from PHY_001_024

with hourly temporal resolution. SMORS takes medium resolution data from PHY_001_024 and uses the SDD techniques to calculate all variables on a high-resolution mesh. The computational mesh for both versions of SMORS have a $1/24^{th}$ degree resolution and hence they quadruple the number of nodes of the original CMEMS grid in the horizontal dimensions. The meshes are aligned in such a way that one out of four nodes in the high-resolution grid is shared with the medium-resolution one. Both versions of SMORS can work operationally 24/7, and provide the same temporal resolution and length of forecast

as the parent medium-resolution model. As SMORS is an operational model, it polls periodically the Copernicus server using Copernicus MOTU library for Python, until the new daily forecasting data are available. Once new data are found, they are automatically downloaded into the local server.

A flowchart representing the workflow of the operational SMORS model is shown in Fig. 5. The process requires registration at CMEMS website and a stable Internet connection to CMEMS servers. All SMORS processing is carried out on a middle spec PC under WINDOWS operating system.

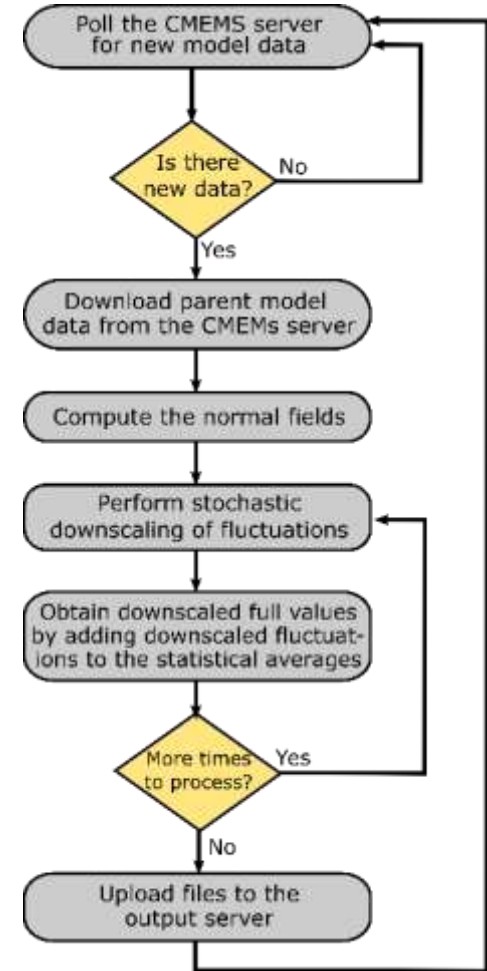

**Figure 5: Flowchart showing the workflow of SMORS operational model.**

The SMORS model uses the correlation function given by Eq. (7) with the parameters similar to those used in creating the Black Sea climatology (Belokopytov, 2018) with the horizontal resolution of 10'×15' which is similar to the resolution of PHY_001_024 model. To perform the downscaling, the SDD method calculates the weighting coefficients $p_i$ for each fine-mesh node using the system of equations (5). The correlation matrices $R_{ij}$ are symmetric, positive-definite but somewhat ill-conditioned, i.e. have condition numbers $CN$ within a few orders of magnitude or larger than the inverse of the machine epsilon. Typical $CN$ numbers for matrices $R$ used in SMORS are in the range of $10^4 - 10^6$ depending on where the point $r_0$ is located, which would be ill-conditioned for single precision arithmetic (inverse epsilon $\sim 1.6 \cdot 10^7$). However despite having the CN

number larger than 1, the matrices with CN numbers in the range of $10^4$ —$10^6$ are easily dealt with by modern computers, and hence can be considered fairly well conditioned for the double precision accuracy (64-bit) used in the computations (inverse epsilon $\sim 10^{16}$).

The numerical difficulties in using ill-conditioned correlation matrices can be reduced by applying advanced numerical
methods, for example Tikhonov method of variational regularisation (Tikhonov,1963), see also (Reichel and Yu, 2015) The detailed algorithm of regularisation with practical examples is described in (Ryabov et al, 2018). After regularisation, a standard method can be used for solving Eq. (5). In case of SMORS, the solution of Eq. (5) does not result in any significant loss of accuracy if all computations are performed with 64-bit precision.

Solving Eq. (5) for all $p_i$ and for every node in the high-resolution 3D grid is a computationally demanding task and requires
the use of finely efficient algorithms. For solving these equations, we have tried three methods: Gaussian elimination, Cholesky decomposition and the conjugate gradient. We have found that the latter is the best choice in terms of speed and numerical stability, if a suitable initial guess is provided, even if no special preconditioner is used. Since $R$ has values equal to one in the diagonal, it can be considered as having a diagonal preconditioner. For the conjugate gradient solver, we have used the code provided by Eigen C++ library (Guennebaud el al., 2020).

For a regular medium grid, the weighting coefficients only depend on the geometry of the grid and the correlation function, therefore they could be computed just once, in advance. The geometry of the grid around fine-mesh points varies significantly in many areas of the Red Sea due to finely variable bathymetry, multiple small islands, and a convoluted coastline. For the initial guess in the iteration process, we use the solution found for the previously considered node (cut to size or padded with zeros if the number of nodes is different). The previously considered node means the nearest adjacent node already solved for.
This approach takes advantage of the fact that both the medium and fine grids are structured and therefore, in most cases, the weights for the neighbouring nodes are similar in value. With this approach, Eq. (5) can be solved for the majority of points on a fine mesh in just a few iterations. Fig. 6 shows the spread of weighting coefficients against distance between points $r_0$ and $r_i$.

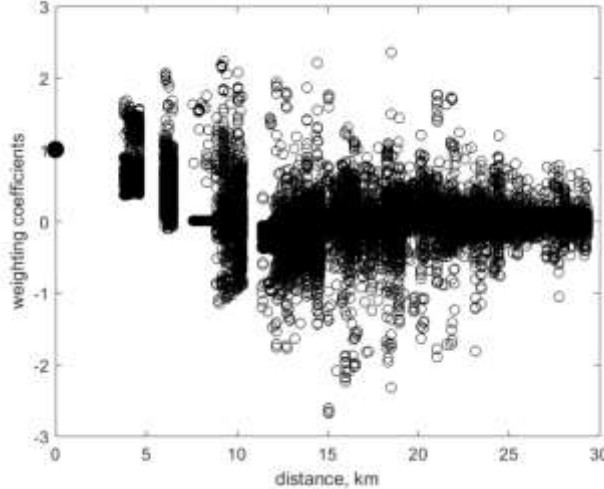

**Figure 6: Distribution of the weights $p_i$ against the distance $\|r_0 - r_i\|$. They are computed between the node on the fine mesh $r_0$ and those points on the medium mesh which are used for calculation of the correlation matrix $r_i$. The plot includes approximately 2.5 million weights calculated for all the fine mesh nodes at the surface of the Red Sea.**

After the weighting coefficients $p_i$ for each fine-mesh node have been found, the downscaling calculation for each fine-mesh node for each parameter at each time point requires a minimum of $2n$ floating point operations, see Eq (1), where $n$ is the number of surrounding medium-mesh points considered for use in downscaling. However, the most time consuming part of calculation is not the calculation of high-resolution values according to Eq (1) but the calculation of weighting coefficients $p_i$ from the system of equations (5) as described above.

## 3 Results

### 3.1 Model Validation

Many deterministic high-resolution models, both in oceanography and meteorology, are prone to errors caused by the so called 'double penalty' issue. The result of this issue is that higher resolution models have a larger root-mean-square-error (RMSE) than lower resolution models (Gilleland et al. 2009). The following quote from (Crocker et al. 2020) explains the situation in more detail. 'One of the issues faced when assessing fine-resolution models against lower-resolution models over the same domain is that often the coarser model appears to perform at least equivalently or better when using typical verification metrics such as root mean squared error (RMSE) or mean error, which is a measure of the bias. Whereas a higher-resolution model has the ability and requirement to forecast greater variation, detail and extremes, a coarser model cannot resolve the detail and will, by its nature, produce smoother features with less variation resulting in smaller errors. This can lead to the situation that despite the higher-resolution model looking more realistic it may verify worse (e.g. Mass et al., 2002; Tonani et al., 2019).

This is particularly the case when assessing forecast models categorically. This effect is more prevalent in finer-resolution models due to their ability to, at least, partially resolve smaller-scale features of interest.' Therefore the 'double penalty' is related to the phenomenon where a model that predicts some spatial feature, but slightly shifted, gets a worse RMSE than a coarser model that completely fails to predict that feature. The physics of the double penalty issue has been studied in detail in (Zingerlea and Nurmib, 2008; ECMWF,2020; Haben et al., 2014). They state, in relation to meteorological forecasts, that 'High-resolution NWP models commonly produce forecasts with seemingly realistic small-scale patterns that can be somewhat misplaced. Traditional point matching verification measures (e.g. the root mean square error, RMSE) would penalize such misplacements very heavily. This penalization actually occurs twice, first, for not having the pattern where it should be, and second, for having a pattern where there should not be one. To the contrary, in the SDD method, the high-resolution output is nudged to the parent model. The SDD method honours the data on the parent coarse grid and hence the spatial structure is anchored onto the coarse grid, therefore there is no additional spatial shift, hence the 'double penalty' error is less likely.

The quality of SMORS has been assessed in two ways. First, the SMORS model output was validated by comparing the model outputs with in-situ observations from ARGO floats (Coriolis, 2020) and sea-surface temperature from the Operational Sea Surface Temperature and Sea Ice Analysis (OSTIA, 2020). OSTIA uses satellite data from a number of sensors as well as in-situ data from drifting and moored buoys. Validation routine follows the guidance produced by GODAE Ocean View consortium (2020)

As an example, Fig. 7 shows the domain-averaged monthly bias and RMSE of SST between PHY_001_024 and OSTIA as well as between SMORS_3D and OSTIA for the year 2016. Both models show a very similar skill with the bias between 0.01 ºC and 0.31 ºC, and the RMSE between 0.37 ºC and 0.68 ºC depending on the month. The differences in errors produced by SMORS and the parent model are very small and their maximum values do not exceed 0.02 ºC, both for bias and RMSE. Therefore, no 'double penalty' issue is seen here.

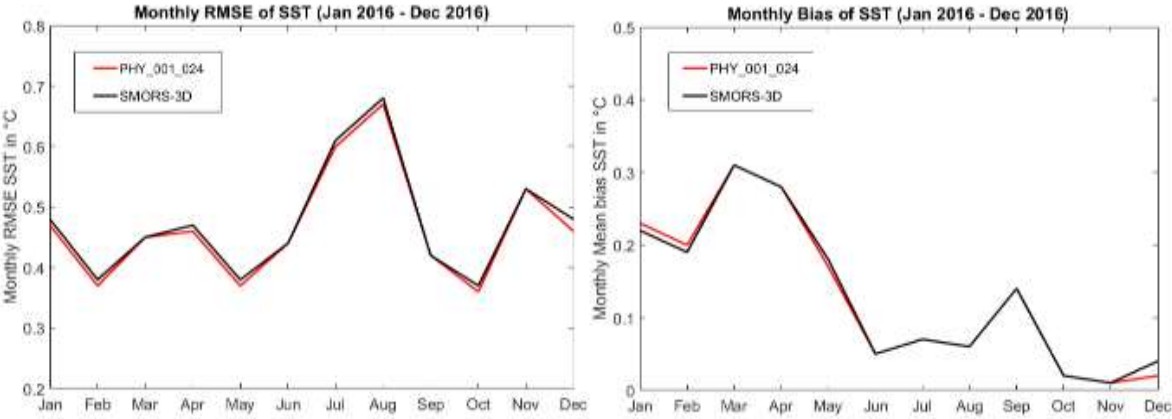

**Figure 7: Monthly RMSE and bias comparisons: Monthly RMSE (left panel) and monthly bias (right panel) between PHY_001_024 and OSTIA (red line); and between SMORS and OSTIA (black line).**

Figure 8 shows comparisons of annual bias (A) and RMSE (C) in temperature between PHY_001_024 and ARGO profiles and between SMORS-3D and ARGO. ARGO floats are constantly moving, for this reason, each ARGO profile is compared to the closest node in each model, similar to the method used in (Delrosso et al, 2016). We use the method for model validation as in the EU My Ocean project (Delrosso et al., 2016) for compatibility reasons. Since SMORS-3D has four times more computational nodes than PHY_001_024, the closest node to the ARGO float can be different for each model. Biases and

RMSE's are then computed for the discrepancies between the models and all the ARGO profiles located inside the Red Sea for the year 2016. At every depth level, only the profiles that include measurements at that depth are included in the calculations.

Again, PHY_001_024 and SMORS-3D show a very similar skill. The discrepancies in temperature between the models and observations are practically the same for both SMORS and PHY_001_024, the biases range between -0.01 °C and 0.71 °C and

the RMSE's are between 0.01 °C and 0.75 °C, depending on the depth. Figure 8 (**B**) shows zoomed-in differences in biases between the two models, which are between -0.011 and +0.014 °C, while the differences in RMSE's shown in Fig. 8 (**D**) are between -0.01 and +0.02 °C.

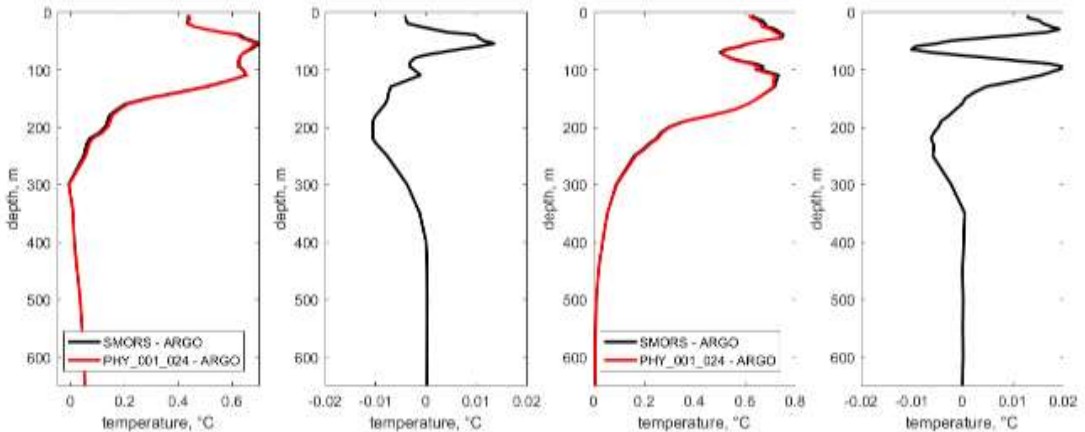

**Figure 8: Annually averaged biases (A) and RMSE's (C) of temperature for the two models: PHY_001_024 – ARGO (dashed line)**
**and SMORS-3D – ARGO (solid line). Plots (B) and (D) show the zoomed-in differences between the lines in plots (A) and (C) respectively.**

The second test was to assess if the SDD method produces noise at high frequencies (in spatial domain). Theoretically, the downscaling onto any existing 'observational' point (in this case a point on the PHY_001_024 mesh) must give exactly the same value as the original data set (Gandin 1963, 1965). Any deviation from this rule is due to computational errors. These

errors were assessed as follows. The output surface data for $u$ and $v$ velocity components from SMORS were subsampled onto the PHY_001_024 mesh and comparison was made by calculating the standard deviation of differences (std_DIF_u, and std_DIF_v). The downscaling was carried out based on daily outputs from PHY_001_024 for each day of the year 2017. Both values, std_DIF_u and std_DIF_v, were very small, of the order $10^{-8}$ m/s, while the typical velocities in the Red Sea were of the order of 0.1—0.2 m/s. Therefore, the potential 'double penalty' error does not occur in the downscaling of profiles.

## 3.2 Eddy and mean kinetic energy

In this section, the results produced by eddy-resolving SMORS-3D model for the year 2017 at the surface are analysed and compared with the eddy-permitting product PHY_001_024 (3D output). The focus of this section is on dynamic properties depending on the currents rather than temperature and salinity, as it is the dynamics where the most significant improvement from downscaling is identified. The Red Sea is known for its mesoscale activity leading to the formation of eddies and filaments, see e.g. (Zhai and Bower, 2013). In order to analyse mesoscale activity, the horizontal velocity $U, V$ is split into slow varying components $\langle U \rangle$, $\langle V \rangle$ representing mean currents and fluctuation components $u$, $v$ representing mesoscale activity:

$$U = \langle U \rangle + u, \quad V = \langle V \rangle + v, \tag{10}$$

where the angle brackets designate statistical mean. As usual, we apply the assumptions of ergodicity and statistical homogeneity of horizontal turbulence generated by mesoscale motions, and for practical purposes we estimate the statistical mean by time averaging for each grid node. The slow varying components are calculated using a low-pass Savitzky-Golay filter of the second order. The cut-off period is taken to be $W = 73$ days as it provides a good separation of fast and slow motion. For each geographical location we calculate the eddy kinetic energy, EKE, and the mean kinetic energy, MKE per unit mass of water as follows

$$\text{MKE} = \frac{1}{2}[\langle U \rangle^2 + \langle V \rangle^2], \quad \text{EKE} = \frac{1}{2}[\langle u^2 \rangle + \langle v^2 \rangle], \tag{11}$$

where slow and fast velocities are defined by Eq (10). In order to assess the degree of separation between slow and fast motions, and the validity of the ergodic assumption, we assess the cross-correlation term $\langle Uu \rangle + \langle Vv \rangle$. Ideally, this term should be zero, as part of the so called Reynolds conditions (Monin and Yaglom, 1971), and hence the following condition must be satisfied:

$$\langle \text{FKE} \rangle = \text{MKE} + \text{EKE}, \tag{12}$$

where

$$\langle \text{FKE} \rangle = \frac{1}{2}[\langle U^2 \rangle + \langle V^2 \rangle],$$

is the time smoothed full kinetic energy, and MKE and EKE are defined by Eq. (11). Fig. 9 shows the time series of area averaged $\langle \text{FKE} \rangle$ and the sum of EKE and MKE. The difference between the curves is small, which confirms the efficient separation between slow and fast motions.

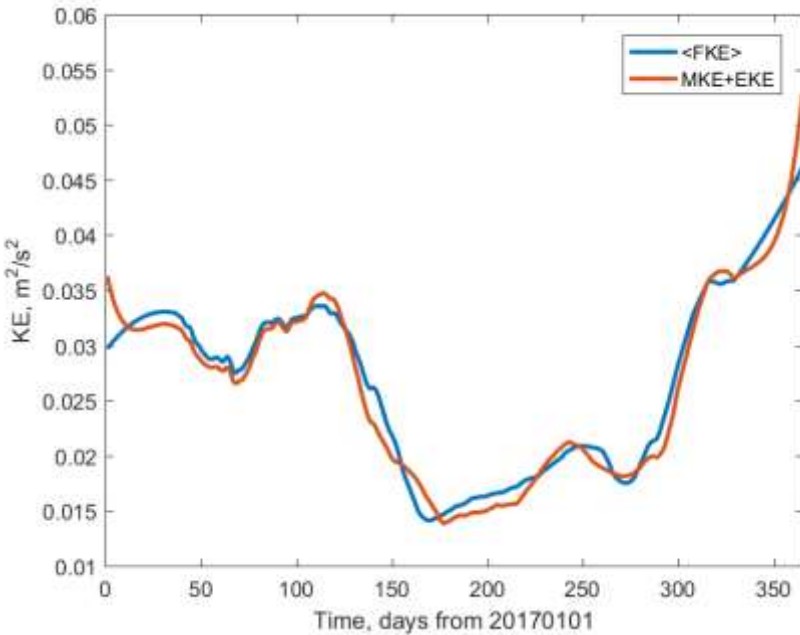

**Figure 9: Temporal variability of area averaged, time smoothed full kinetic energy (blue); and the sum of eddy and mean kinetic energy (orange) in the surface layer of the Red Sea during 2017.**

The time series of EKE and MKE averaged over the entire Red Sea show a relatively small difference between eddy-permitting
(PHY_001_024) and eddy-resolving (SMORS) models. The maximum difference in EKE is only 4.5% of its root-mean-square value for the year 2017, however this ratio is slightly larger at 10.8% for MKE. These differences are discussed in section 4 .

### 3.3 Analysis of vorticity and enstrophy

Another important dynamic characteristic of the ocean circulation is vorticity (Rossby, 1936). Analysis of vorticity has been the basis of much of classical wind-driven ocean circulation theory (Marshall, 1984). The time series of relative vorticity
averaged over the whole Red Sea and calculated from the outputs of the parent model (PHY_001_024) and SMORS is shown in Fig. 10. Values of vorticity calculated at individual grid points from the fine resolution model are typically higher than from the medium-resolution model. Higher values of vorticity are a result of better representation of horizontal gradients in velocity by finer-resolution SMORS model. This effect is seen in both slow and fast varying components of vorticity. The difference in the area-averaged vorticity is a result of differences in the shape of the coastline and a number of islands represented in the
coarse and fine grids.

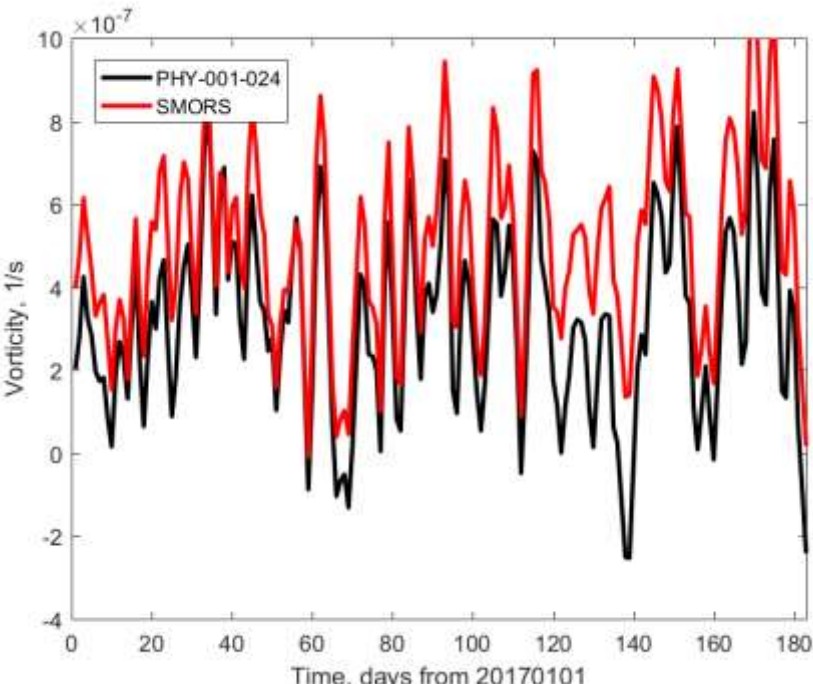

**Figure 10: Area averaged vorticity as a function of time calculated from PHY_001_024 (black) and SMORS (red).**

The difference in vorticity sampled on the coarser PHY_001_024 grid is shown in Fig. 11. The root mean square of the difference (RMS-DV) in vorticity calculated over the entire Red Sea is smaller but comparable with RMS-V of the vorticity itself. In the example shown in Fig. 11, the percentage ratio of the two is as high as 17%. The difference is larger in the areas of intensive mesoscale activity in the central and northern parts of the Red Sea where the coarser PHY_001_024 model shows weaker and smoother velocity gradients.


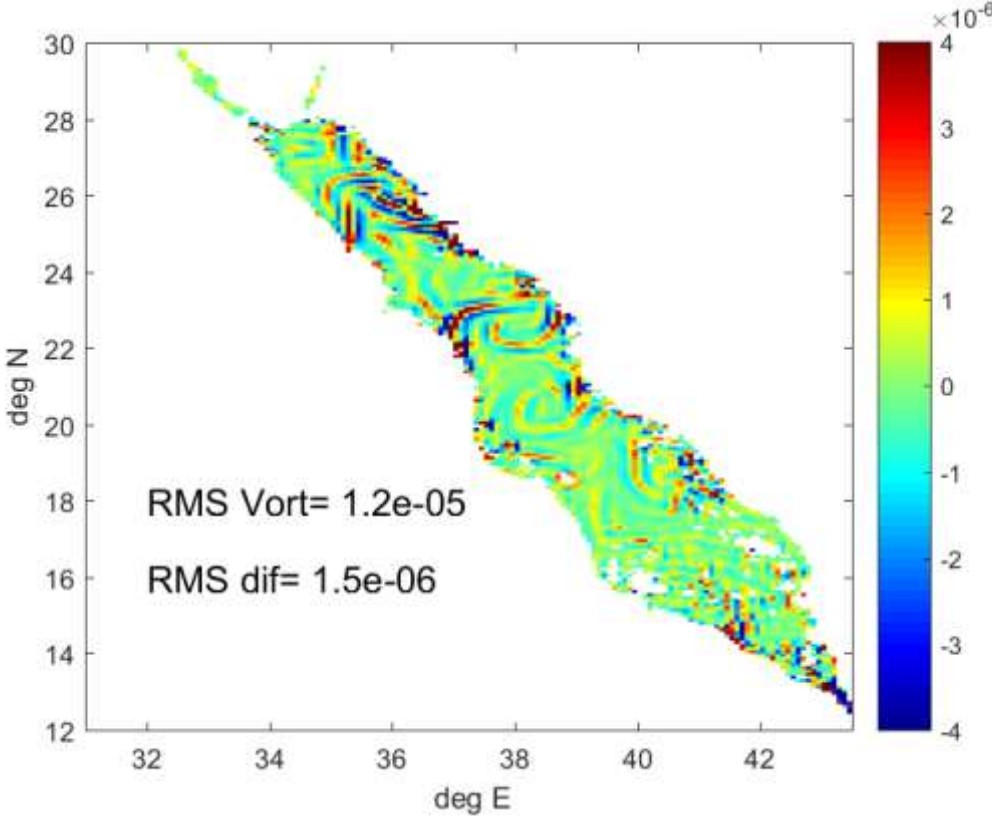

**Figure 11: Snapshot of a difference in the surface current vorticity (s$^{-1}$) calculated from SMORS and PHY_001_024 models for the 1$^{st}$ of April 2017.**


An important dynamic characteristic of the mesoscale activity is the local enstrophy defined as the square of relative vorticity at a location and the total enstrophy defined as an integral of local enstrophy over the horizontal dimensions of a domain

$$\text{Enstr}(t) = \int_{\text{Red Sea}} \|\nabla \times \boldsymbol{U}(x, y, t)\|^2 dA. \tag{13}$$

In the inviscid flow, enstrophy is conserved in a closed system, and hence variation of area averaged (or area integrated)
enstrophy gives an indication of the role of ocean-atmosphere interaction and viscous dissipation (Lesieur, 2008). The value of enstrophy is also indicative of the rate of dissipation of kinetic energy, and hence a correct estimate of enstrophy provides a better insight into the underlying processes of transformation of energy in the basin. The time series of area averaged enstrophy, i.e. integral enstrophy defined by Eq. (13) divided by the area of the domain, is represented in Fig. 12. Enstrophy is minimal in the summer period when mesoscale activity is reduced.


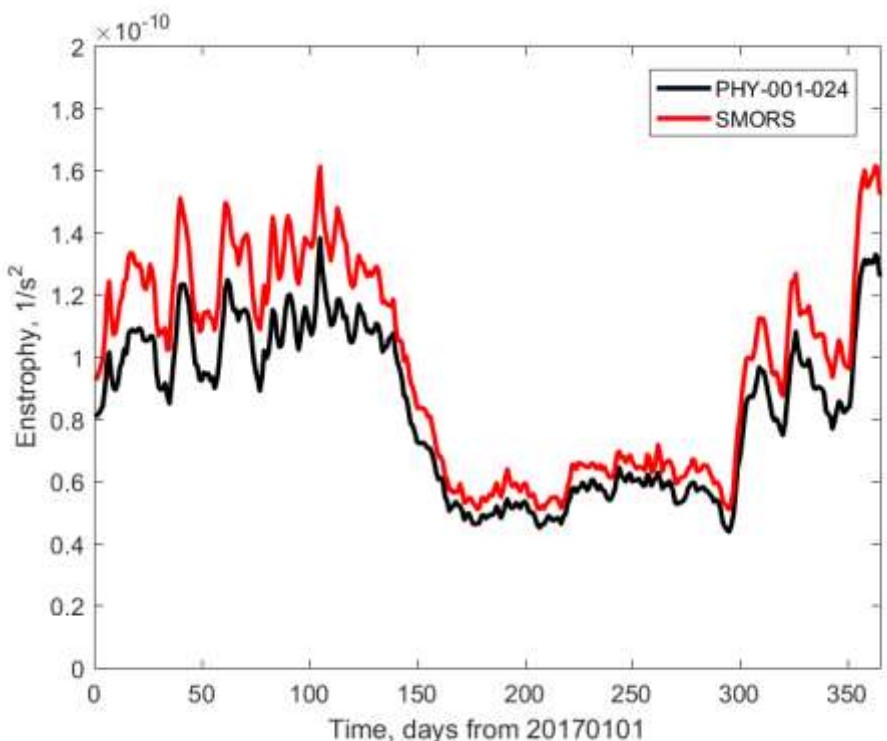

**Figure 12: Seasonal variability area averaged enstrophy assessed from medium resolution PHY_001_024 (black) and fine-resolution SMORS (red) models.**

The spatial distribution of enstrophy produced by SMORS model at the end of the eddy-intensive summer period is shown in

Fig. 13. There are two strong eddies in the central part of the sea, which have been shown to influence the overall mesoscale

dynamics of the sea (Zhai and Bower, 2013)

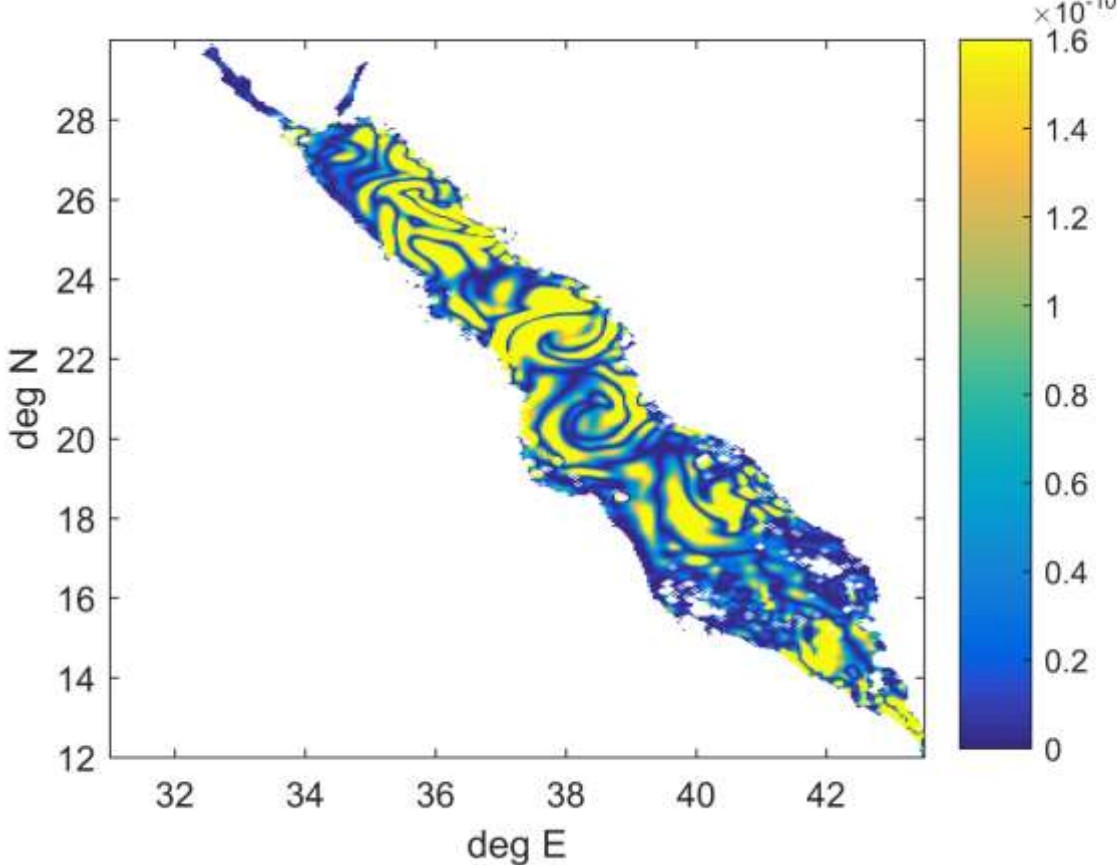

**Figure 13: Distribution of local enstrophy of surface currents (s$^{-2}$) in the Red Sea on the 1$^{st}$ April 2017 estimated from SMORS model output.**

**4 Discussion**

This paper presents an efficient method for fine-resolution ocean modelling based on downscaling from a medium to fine resolution mesh. In contrast to common downscaling methods that rely on solving dynamic equations in a smaller sub-region,
the new method uses a combination of the deterministic and stochastic approaches. The philosophy behind the new method named stochastic-deterministic downscaling, or SDD, is that at smaller scales, not resolved by the parent model, the chaotic, turbulent nature of water motion can be well represented by its statistical properties. The method utilises mathematical tools similar to those developed for optimal interpolation of observations and data assimilation in ocean modelling. The main difference is that instead of assimilating a relatively small number of observations, the SDD method assimilates all the data
produced by a parent model. The novelty of the SDD method in this respect is that the methodology originally developed for assimilating a limited number of observational data is modified and applied to assimilating coarse model data into the fine model. In contrast to common data assimilation methods, the new information comes from the computation of many millions

of downscaling factors, see Eq (5) and Fig. 6, which in turn uses the correlation matrices. Therefore, the SDD method should be treated as experimental at this stage. The SDD approach is first applied to and tested in an idealised case and then applied to create an operational Stochastic Model of the Red Sea, or SMORS, based on data available via Copernicus Marine Environment Monitoring Service. SMORS has a 1/24[th] degree of resolution, compares favourably with observations, and allows to reveal greater granularity of the dynamical features of the Red Sea, in particular those dependent on the shear of ocean currents.

The statistical links used by the SDD method can be interpreted in a way similar to the theory of fully developed turbulence. According to the Kolmogorov's law the statistically uniform and isotropic (in 3D) turbulence can be described by a universal power density spectrum (the law of 5/3) which is equivalent to the law of 2/3 for the structure functions (Gandin and Kagan, 1976). The studies of velocity fluctuation in the upper air showed that the correlation function for geopotential heights has a universal shape for distances large enough to consider the processes to be 2-dimensional (in the horizontal) but smaller than the Rossby radius of deformation (Yudin, 1961). Previous studies confirmed that the small-scale velocity fluctuations in well-developed turbulence exhibit universal scaling properties independent of the large-scale flow structures (Nelkin, 1994).

The correct identification of the correlation function and, in particular, its digital representation in the form of the correlation matrix $R$ given by Eq. (6) is critical for the success of the SDD method. Theoretically, matrix $R$ should be symmetric and positive-definite, however this is not always the case when the matrix is derived from observations (Tabeart et al., 2020). There are a number of ways to estimate the numerical values of elements in the correlation matrix (6), see e.g. (Park and Xu, 2018; Fu et al., 2004). For the purpose of downscaling, an optimal design of matrix $R$ would ideally reflect the structures at a short range, comparable with the resolution of the parent model. It has been shown that the dependence of the autocorrelation matrix on the horizontal distance $\|r_i - r_j\|$ is universal at small separations and is close to universal at separation comparable to Rossby radius of deformation (Yudin,1961, Gandin, 1963). This is consistent with a general view that the Rossby radius is a predominant scale for coherent structures in the ocean such as mesoscale eddies, which are typically 2—3 times larger than the first baroclinic Rossby radius, see e.g. (Barenblatt, 1992; Beron-Vera et al, 2019; Badin et al, 2009).

Whilst the elements of the correlation matrix $R_{ij}$ depend only on the distance between the contributing points as specified by Eq (7), the weighting coefficients $p_i$ are not a unique function of the distance between the points $r_0$ and $r_i$. This means that standard interpolation methods such as bi-linear, polynomial, inverse distance etc. based on a fixed dependence of weights on distance cannot be used as an adequate substitution to the method described above, as this method minimises the error between the true and estimated values on the fine mesh (Gandin, 1965) and hence it gives the best possible estimates of ocean parameters. The distribution of weights is generally different for different points $r_0$ on the fine mesh, however it may be the same for a subset of points away from the coastlines due to the regular structure of the medium mesh.

In theory, in order to solve the $N_{\text{fine}}$ systems of equations (5) for each node on the fine mesh, the matrix $R_{ij}$ has to involve all the nodes of the medium mesh in the domain, because eq. (7) gives values of $R_{ij}$ different from zero, no matter what is the distance between nodes. In practice, this is undesirable. Firstly, large systems of equations require vast computational resources

to be solved. Secondly, large correlation matrices are known to have large condition numbers (Tabeart et al., 2020) and this problem gets worse as the matrix size increases. Thirdly, the data from nodes which are away by more than a few Rossby radii, are not physically correlated to small scale variations within a single grid cell of the medium mesh.

The reason for matrix $R$ to be ill-conditioned is that whilst its largest elements (equal to 1) are on its diagonal, there are many non-diagonal elements which have similar, however slightly smaller values. This is because the grid cell on the medium mesh is smaller than a typical size of mesoscale features (2—3 times the Rossby radius). The Rossby radius determines the scale of coherency of ocean parameters, therefore the correlations between neighbouring points on the medium mesh are close to 1. The baroclinic Rossby radius in the Red Sea is about 10—30 km (Manasrah, 2006; Zhai and Bower, 2013). In principle, the matrix $R$ could have been made more diagonally dominant and its condition number would have reduced if using a coarser mesh. However, this would have led to the loss of statistical information at smaller scales and hence would introduce larger errors in the downscaling process.

The SMORS model is a computationally efficient way to generate finer-resolution 3D oceanographic forecast for the Red Sea. With all considerations listed above, the whole process of downloading the file from the Copernicus server, finding the weighting coefficients, downscaling the fields of U, V, T and S, and saving the output NetCDF file, takes about three hours on a single core of a typical desktop PC. Efficiency of SMORS model is seen from the following comparison. The time required for both SMORS-3D and SMORS-2D to run on a desktop PC with a single core is comparable to the time required for a purely deterministic model (such as NEMO) with the same resolution to run on a HPC cluster with 96 computing cores. If faster speeds for the SDD method are needed, the algorithm is parallelizable on a modern desktop PC or, of course, on an HPC cluster. The running of the model can be further optimised by applying the SDD method only to a selection of depth levels used by the parent model, either horizontal or curved.

The SDD method was tested using an idealised case where the true solution is known (sub-section 2.2). The SDD method showed good ability to recover smaller scale details of mesoscale eddies which were missed by the parent eddy-permitting model, as well as fine-resolution interpolating models based on a prescribed analytical formula for weighting coefficient. The comparison of the maps and transects produced by the parent (coarse), analytical interpolating and SDD based models, see sub-section 'idealised case' above, shows the benefits of the SDD method in comparison with downscaling approaches based on analytical interpolation routines. The SDD method produces data on the fine mesh which are much closer to the true solution than simple bi-linear or bi-cubic interpolation. In contrast to the analytical interpolation methods which smooth the gradients, the SDD is capable of recovering sharp gradients and details of the ocean fronts. Similar qualities are seen in the real world application of the SDD to the Red Sea, where SMORS model shows finer granularity of the velocity, vorticity and enstrophy fields, than the parent model. The enstrophy field has larger values on the fine than the coarse grid. This probably relates to the additional power in the Fourier spectrum caused by the fact that the derivatives for the fine grid are calculated using smaller grid spacing than on the coarse grid and hence have a potential to show sharper gradients.

The high-resolution SMORS model not only provides a greater granularity of the spatial distribution that a coarser parent model misses, it also gives different estimates for the area averaged ocean variables. For example, while the differences between the full kinetic energy computed with the parent and fine-mesh models shown in Fig. 9 are relatively small, they can be attributed to the ability of SMORS model to reveal local extrema in velocity which are not resolved by PHY_001_024. The seasonal variation of EKE and MKE produced by SMORS shows less mesoscale activity in summer and more in winter, similar to the results obtained with high-resolution deterministic model MITgcm (Zhan et al, 2016). Conceivably the statistics are related to those structures which determine the correlation matrix, however is there is also a deterministic element in the small-scale as the SDD honours the data from the parent deterministic model and in the seasonal variation of the climatic 'norms' against which the fluctuations are calculated.

The knowledge of the structure and evolution of vorticity field in the ocean provides vital information about ocean circulation. For example, the effect of mesoscale eddies is to produce a transport of vorticity from regions of high to regions of low vorticity (Corre et al, 2020). Mesoscale flows are the primary cause for the ocean transport of heat, carbon and nutrients (Robinson, 1983). Furthermore, the sub-mesoscales (1 km—10 km) are emerging as an important dynamical regime. Dynamical processes at the mesoscales and sub-mesoscales are relevant for understanding and modelling interactions near the coasts and the movement of ocean heat under high latitude ice-shelves that can have important implications for sea level (GFDL, 2020). The knowledge of vorticity values helps assess the stability of the 2D flow to 3D instabilities (Flor, 2010). Therefore, its accurate calculation is a desired quality of any ocean circulation model.

Vorticity is closely linked to another important feature of the flow, its local enstrophy. The SMORS model reveals high level of granularity in enstrophy distribution in particular north and south of the persistent eddies in the central part of the Red Sea as shown in Fig. 14 which demonstrates the spatial distribution of differences in enstrophy computed by SMORS and PHY_001_024. The difference in vorticity and enstrophy is a result of the fact that the velocity gradients in the fine-resolution child model are sharper than in the parent model. This effect is clearly seen in the idealised case, where the true solution was known (see sub-section 2.2 'Idealised case'). Hence, the enstrophy is nearly always greater in the fine-resolution model. The difference between the models can be characterised by the ratio of root-mean-square of difference in enstrophy to the root-mean-square of enstrophy itself which is as much as 21% for the snapshot shown in Fig. 14.

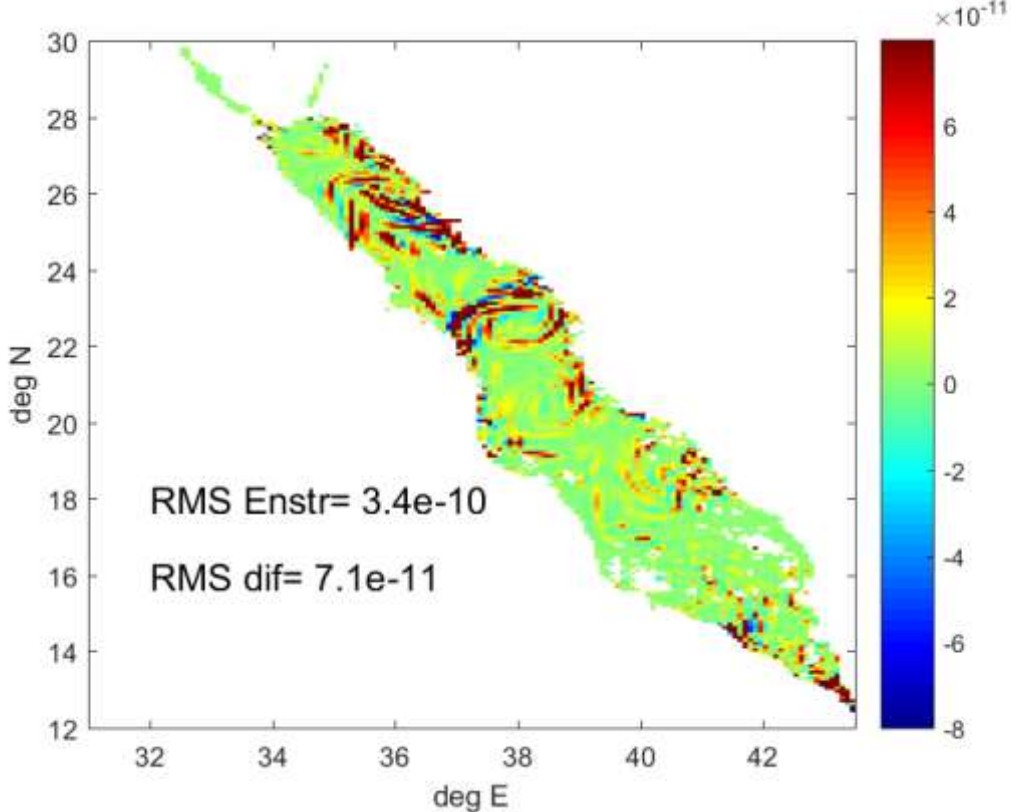

**Figure 14: Spatial distribution of differences in surface enstrophy ($s^{-2}$) between the fine-resolution and medium-resolution models on the 1st of April of 2017.**

The benefits of the higher-resolution model are better seen in a zoomed-in area shown in Fig. 15. The fine-resolution model provides better granularity and it also better resolves the maxima in enstrophy which were not resolved by the parent medium resolution model.

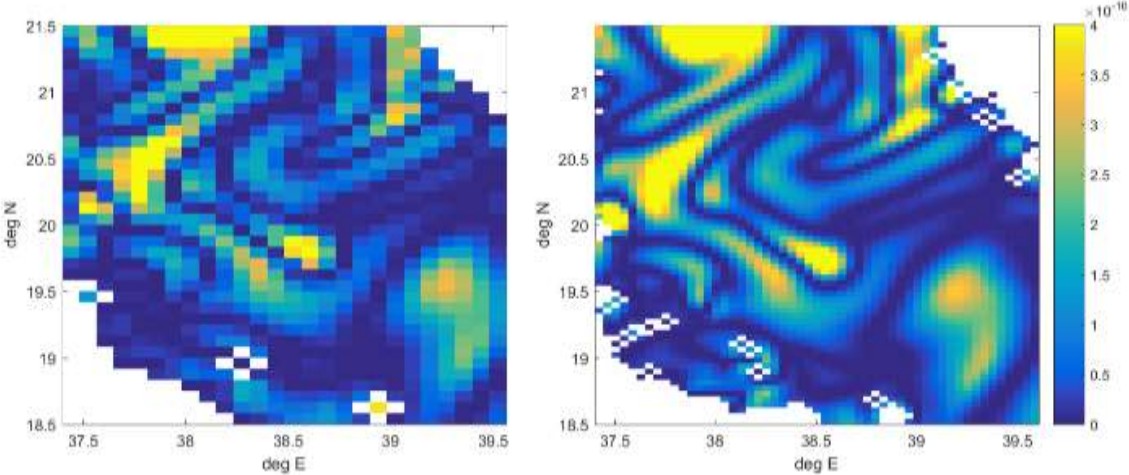

**Figure 15: Enstrophy of surface currents in the central part of the Red Sea presented by eddy-permitting PHY_000_024 (left) and eddy resolving SMORS (right) model: The white areas represent land and are different as the fine model uses higher resolution bathymetry and coastline, which reveals more small islands.**

## 5 Conclusion

We present an efficient method for high-resolution ocean modelling which uses downscaling from a medium resolution model
and is based on the combination of the deterministic and stochastic approaches. We call this method stochastic-deterministic downscaling, or SDD. The philosophy behind SDD is that at smaller scales the chaotic, turbulent nature of water motion can be represented more efficiently by incorporating methodologies commonly used in the study of turbulence. The method utilises the same mathematical tools which were originally developed for objective analysis of observational data in meteorology and then for data assimilation in ocean modelling. The main difference is that instead of assimilating a relatively small number of
observations, the SDD method assimilates a vast number of gridded data produced by a parent model. The SDD model has the same length of forecast, vertical discretisation and frequency of outputs as the parent model. The method can be applied to individual depth levels independently. We believe that the methodology behind the SDD method, i.e. data assimilation from coarser models rather than from observations, can be extended for the use of other data assimilation techniques, such as 3D-VAR, Kalman filtering etc. We also think that a combined data assimilation from observations and coarser models is also
possible.

The validation of SDD in an idealised setting, where the exact solution is known, demonstrates its ability to reconstruct finer-scale features which are lost in the parent lower resolution model. The method is shown to be efficient in case where the parent model is eddy-permitting, while the downscaled model is eddy-resolving. The SDD type model named SMORS (Stochastic Model of the Red Sea) was set up for the Red Sea with a resolution of 1/24° using a parent model from Copernicus Marine
Environmental Service with 1/12° resolution and ran operationally for more than a year. Validation against the parent model,

in-situ and satellite observations confirmed that SDD is not prone to generating additional errors due to the 'double-penalty' effect which is common for purely deterministic fine-resolution models.

The fact that SDD is well suited to downscaling of noisy (stochastic) data makes such a method more attractive than it would otherwise be. The cost of increased resolution using a fine-resolution deterministic model is so high that such a method is worthwhile, providing both increased resolution in the simulation and low additional noise.

The SMORS model uses advanced numerical algorithms, is computationally efficient and can be run on a single core of a desktop PC operationally. The running of the model can be further optimised by applying the SDD method only to a selection of depth levels used by the parent model, either horizontal or curved. It is likely that the method could be further developed by incorporating more complex data assimilation schemes.

## Author contribution

GS conceptualised and designed the study, performed the analysis, and drafted the manuscript.

JO created numerical schemes, carried out software development, contributed to the analysis and writing the manuscript. VB contributed to the development of the algorithm and selection of appropriate parameters.

## Code/Data availability

The data generated by the parent model is available via Copernicus Marine Environment Monitoring Service (CMEMS 2020), product ID PHY_000_024.

## Competing interests

The authors declare that they have no conflict of interest.

## Acknowledgment

Funding for this study was provided by the University of Plymouth Enterprise LTD. The authors are thankful to Xavier Francis for his help in the validation of the earlier version of SMORS.

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
