# Peer review of "High resolution stochastic downscaling method for ocean forecasting models and its application to the Red Sea dynamics"

_Ocean Science, 2020_

## Referee Comment (RC1)

Review of "High resolution stochastic downscaling method for ocean forecasting models and its application to the Red Sea dynamics" by Georgy I. Shapiro, Jose M. Gonzalez-Ondina, Vladimir N. Belokopytov

**Overall comments**

As I read it, the main idea in this paper is that optimal interpolation (OI) is the best method for interpolating a field $f$ of values specified on a set of regular grid points $x_i$, to another set of points. This may seem obvious from the name "optimal interpolation", but OI is not usually thought of in this context. It's usually thought of as a method for combining a number of irregularly spaced noisy observations with a climatology or model background to produce an optimal analysis. Figure 2 of the paper shows that, for some regularly gridded fields, OI provides much better interpolation results than standard methods like bi-linear or bi-cubic interpolation. If this is correct, which I think it is, it could be important for several reasons. First, in the data assimilation context, interpolating the model to the observations accurately is widely acknowledged to be a key step. So that doing that more accurately should improve the results. Second, as shown in figure 14, maps of fields with a lot of fine scale structure, such as the vorticity, may be rendered with greater fidelity using OI for interpolation rather than other interpolation methods.

The paper suggests (line 413) that the SMORS method should be treated as experimental at this stage. It seems to me that this is correct for a couple of reasons. First the types of fields for which OI provides significantly better results than cubic bi-linear interpolation needs further clarification. Figure 2 uses an example in which the grid barely resolves the fields in the x-direction. It may be that it is only in such cases that OI gives significantly better results. Are the authors able to explore that in a little more detail? Second the model field that is being interpolated should not usually be regarded as being precisely correct. More specifically an ocean (or atmosphere) model is well known to be unreliable near the grid-scale. Fields are typically either noisy at the grid-scale or overly damped. It is also a moot point whether the fields represent point values or grid cells means. To what extent it is possible to extract more information near the grid-scale from model fields by using OI needs more investigation. The fact that OI is well suited to interpolation of noisy (stochastic) data makes such an investigation more attractive than it would otherwise be. The cost of increased resolution is so high that such an investigation is worthwhile, though the value of increased resolution is not necessarily in the increased detail in the simulations.

I think this paper is interesting and is likely to be suitable for publication after significant revision.

**Specific comments**

Title: I wonder whether the words Optimal interpolation (or objective analysis?) should be in the title. For example: "Extraction of near grid-scale dynamical information from model fields using optimal interpolation." I'm a bit concerned that the main point of the paper is not evident from the title.

Abstract: The abstract describes the sort of problem that is being addressed and does introduce the principal idea in lines 20-22. The important point about the lack of a double penalty is clearly made on lines 14 and 27. So the abstract is a reasonably informative summary of the paper.

Lines 20-21: I found the sentence starting on line 20 somewhat difficult to follow.

**Introduction:**

Lines 54 and 55: it might be worth mentioning that the commonly used, more "modern", variational methods are also closely related to OI (Lorenc 1986, QJRMS).

The literature on methods for post-processing of model outputs using Kalman filters should probably be discussed in the introduction. I think the main idea being pursued in this paper is somewhat different from the main ideas in that literature but the techniques are clearly related.

One might ask whether the method proposed is a post-processing of model output or a statistical model in its own right. It is described both as a Statistical Model (in SMORS) and a Stochastic Deterministic Downscaling (SDD) method. Personally I would view it as a post-processing method but do not feel strongly about this semantic issue.

The introduction does introduce relevant material but at the end of it one does not have much more insight into how the proposed technique works. The structure of the paper is not described.

**Section 2.1**

Lines 107-109: The primed quantities (that are interpolated) are deviations from monthly means for the Red Sea. These values would not normally be available in real-time. I imagine that deviations from climatology would be a satisfactory alternative.

Line 133: The use of Gaussian and SOAR functions for the autocorrelation function dates back well before Fu et al (2004). In data assimilation the difficulties / uncertainties in the calculation of this function are usually emphasised quite strongly. The textbook by R. Daley (Atmospheric Data Analysis 1991) is a good source of information on the techniques discussed in this paper. Lorenc 1981 QJRMS describes a fairly sophisticated method for retaining consistency of solutions between points.

**Section 2.2**

The idealised case has a=4.1km and the parent grid has $\Delta x = 10$ km. So the sinusoidally varying field in one direction is really close to the 2-grid point wavelength. It is very impressive how well the OI solution handles this problem (Figure 3 of the paper and line 201 -204). A brief summary of results for some less extreme interpolation cases would probably be informative.

**Section 2.4**

Figure 6: The lack of a double penalty is certainly an interesting result and the comparison with OSTIA data seems sound to me (though I'm not an expert in this issue). It's not clear to me what explains the lack of a double penalty. Is the model SST a relatively smooth field in which case OI and linear interpolation might give relatively small differences? I think there needs to be some further quantification of the double penalty. For example, one could calculate the rms of f' at all points on the high resolution grid that do not coincide with low resolution grid points for the OI, bi-linear and bi-cubic fields. How do these rms values compare with those in figure 6? One might also ask how these rms values compare with the rms of the values at the original (lower resolution) gridpoints. This calculations would help to shed some light on the lack of a double penalty.

Lines 296-298: It seems strange to use nearest neighbour values in the ARGO inter-comparison. With the OI method one can do much better interpolations! Some readers may be concerned that the nearest neighbour method could somehow account for the lack of a double penalty (see previous paragraph).

Lines 311-318: This point that the interpolation will reproduce the field exactly at the parent grid points (to within truncation errors) is an important one. Many readers would find it helpful to mention this earlier in section 2.1.

Section 3: Results

Some results have already been presented in section 2. So the section title seems strange. Results for vorticity or Vorticity diagnostics would be a better title.

Line 353: describing the two models as eddy-permitting and eddy-resolving seems contentious at this point.

Line 360: Could you confirm that Figure 9 shows area mean values of vorticity? Lines 363-364 suggest it does. The phrase "Absolute values of vorticity" in line 360 gives the reader some cause for uncertainty on this point. I suggest you remove "Absolute" from that phrase.

Figure 14 is a nice illustration of the potential of this method. I wonder whether there are any plotting packages which use this type of approach.

---

## Author Response (AR1)

**Responses to reviewer 1 comments on the manuscript 'High resolution stochastic downscaling method for ocean forecasting models and its application to the Red Sea dynamics'**

The authors are thankful to the reviewer for thorough and helpful comments. Our responses are given below. References to line numbers are for the amended MS.

**Comment.** Figure 2 uses an example in which the grid barely resolves the fields in the x-direction. It may be that it is only in such cases that OI gives significantly better results. Are the authors able to explore that in a little more detail?

**Response.** It is correct that the maximum enhancement produced by SDD compared to simple interpolation is expected when the parent model barely resolves the field. If the ocean feature is well resolved by the parent model, there is no need for further refinement. For example, if the zonal size of the eddy is increased to 40 km instead of 13 km, and hence it is reasonably well resolved by the parent model with $\Delta x=10km$, then the RMSE produced by SDD is similar to that of bi-cubical interpolation. On the other hand, if the parent model misses the features completely, e.g. no eddy permitting, then the SDD method does not have enough information to re-create the smaller scale features. The clarification is given in lines 215-220.

**Comment.** Second the model field that is being interpolated should not usually be regarded as being precisely correct. More specifically an ocean (or atmosphere) model is well known to be unreliable near the grid-scale. Fields are typically either noisy at the grid-scale or overly damped. It is also a moot point whether the fields represent point values or grid cells means. To what extent it is possible to extract more information near the grid-scale from model fields by using OI needs more investigation. The fact that OI is well suited to interpolation of noisy (stochastic) data makes such an investigation more attractive than it would otherwise be. The cost of increased resolution is so high that such an investigation is worthwhile, though the value of increased resolution is not necessarily in the increased detail in the simulations.

**Response**. This is correct, the SDD method works better than simple interpolation in a noisy situation. We have added a small subsection 'Effect of noise in the input data' and a table showing the errors produced by different methods when the parent model outputs are noisy. This also shows that the coarse field does not need to be regarded as precisely correct. We are thankful to the reviewer for this comment and suggestion to discuss the treatment of noisy data in the MS.

**Comment.** Abstract: The abstract describes the sort of problem that is being addressed and does introduce the principal idea in lines 20-22. The important point about the lack of a double penalty is clearly made on lines 14 and 27. So the abstract is a reasonably informative summary of the paper.

**Response.** Thank you.

**Comment.** Lines 20-21: I found the sentence starting on line 20 somewhat difficult to follow.

**Response.** The sentence is now re-worded. (Line 20)

**Comment.** Lines 54 and 55: it might be worth mentioning that the commonly used, more "modern", variational methods are also closely related to OI (Lorenc 1986, QJRMS).

**Response.** The reviewer's suggestion is implemented in lines 56-57.

**Comment**. The literature on methods for post-processing of model outputs using Kalman filters should probably be discussed in the introduction. I think the main idea being pursued in this paper is somewhat different from the main ideas in that literature but the techniques are clearly related**.**

**Response.** The reference to Kalman filtering and other methods are briefly given in lines 47-48 and 56-57. The reference to work by Lorenc (1986) is given where it is shown that the Kalman filter and more modern variational methods are closely linked to the original OI and they can be described using a common Bayes analysis framework.

**Comment.** One might ask whether the method proposed is a post-processing of model output or a statistical model in its own right. It is described both as a Statistical Model (in SMORS) and a Stochastic Deterministic Downscaling (SDD) method. Personally I would view it as a post-processing method but do not feel strongly about this semantic issue.

**Response.** We prefer to term the SDD method as part of the model based on how it is implemented in the code. This is shown in the flowchart in Fig.4.

**Comment.** The introduction does introduce relevant material but at the end of it one does not have much more insight into how the proposed technique works. The structure of the paper is not described.

**Response.** In the amended MS the structure of the paper is now explained in lines 82-86. The insight into how the proposed technique (SDD method) works is given in lines 66-73.

**Comment.** Lines 107-109: The primed quantities (that are interpolated) are deviations from monthly means for the Red Sea. These values would not normally be available in real-time. I imagine that deviations from climatology would be a satisfactory alternative.

**Response.** This is correct. Clarification is given in lines 115-116.

**Comment.** Line 133: The use of Gaussian and SOAR functions for the autocorrelation function dates back well before Fu et al (2004). In data assimilation the difficulties / uncertainties in the calculation of this function are usually emphasised quite strongly. The textbook by R. Daley (Atmospheric Data Analysis 1991) is a good source of information on the techniques discussed in this paper. Lorenc 1981 QJRMS describes a fairly sophisticated method for retaining consistency of solutions between points.

**Response**. We agree. The text is amended as requested and an additional reference to Daley (1991) is added in lines 145-146.

**Comment**. The idealised case has a=4.1km and the parent grid has $\Delta x$ = 10 km. So the sinusoidally varying field in one direction is really close to the 2-grid point wavelength. It is very impressive how well the OI solution handles this problem (Figure 3 of the paper and line 201 -204). A brief summary of results for some less extreme interpolation cases would probably be informative.

**Response**. The efficiency of SDD in comparison to different interpolation method is added as a new sub-section in lines 225-252.

**Comment**. Figure 6: The lack of a double penalty is certainly an interesting result and the comparison with OSTIA data seems sound to me (though I'm not an expert in this issue). It's not clear to me what explains the lack of a double penalty. Is the model SST a relatively smooth field in which case OI and linear interpolation might give relatively small differences? I think there needs to be some further quantification of the double penalty. For example, one could calculate the rms of f' at all points on the high resolution grid that do not coincide with low resolution grid points for the OI, bi-linear and bi-cubic fields. How do these rms values compare with those in figure 6? One might also ask how these rms values compare with the rms of the values at the original (lower resolution) gridpoints. This calculations would help to shed some light on the lack of a double penalty.

**Response**. Double penalty phenomenon is more evident in the high resolution models (Gilleland, E., Ahijevych, D., Brown, B. G., Casati, B. and Ebert, E. E.: Intercomparison of spatial forecast verification methods. Weather Forecast, 24(5), 1416-1430, 2009). The SDD method honours the data on the parent coarse grid and hence the spatial structure is anchored onto the coarse grid, therefore there is no additional spatial shift and no additional double penalty effect compared to the parent model. Clarification is given in lines 337-341 of the revised MS.

**Comment**. Lines 296-298: It seems strange to use nearest neighbour values in the ARGO inter-comparison. With the OI method one can do much better interpolations! Some readers may be concerned that the nearest neighbour method could somehow account for the lack of a double penalty (see previous paragraph).

**Response**. We use the nearest neighbour method for compatibility reasons, as it is used for validation of MyOcean / Copernicus Marine Environment Monitoring Service products, see e.g. Delrosso, D., Clementi,E., Grandi,A., Tonani,M., Oddo, P., Feruzza, G., Pinardi,N. 2016. Towards the Mediterranean forecasting system MyOcean v5: numerical experiments results and validation, 2016. INGV technical report, No 345, ISSN 2039-7941. Clarification and additional reference are given in line 358.

**Comment**. Lines 311-318: This point that the interpolation will reproduce the field exactly at the parent grid points (to within truncation errors) is an important one. Many readers would find it helpful to mention this earlier in section 2.1.

**Response**. We agree. Clarification is added in lines 160-163.

**Comment**. Some results have already been presented in section 2. So the section title seems strange. Results for vorticity or Vorticity diagnostics would be a better title.

**Response**. The results presented in section 2 have now been moved to section 3, and new sub-sections are created: 'Eddy and mean kinetic energy' and 'Analysis of vorticity and enstrophy'.

**Comment**. Line 353: describing the two models as eddy-permitting and eddy-resolving seems contentious at this point.

**Response**. Analysis of the efficiency of the SDD method for eddy-permitting and eddy-resolving models is now added throughout the text.

**Comment**. Line 360: Could you confirm that Figure 9 shows area mean values of vorticity? Lines 363-364 suggest it does. The phrase "Absolute values of vorticity" in line 360 gives the reader some cause for uncertainty on this point. I suggest you remove "Absolute" from that phrase.

**Response**. The text amended as advised (lines 419-420)

**Comment**. Figure 14 is a nice illustration of the potential of this method. I wonder whether there are any plotting packages which use this type of approach.

**Response**. We are not aware of any plotting packages which use this type of approach.

We thank Reviewer 1 for helpful comments

**Responses to reviewer 2 on the manuscript 'High resolution stochastic downscaling method for ocean forecasting models and its application to the Red Sea dynamics'**

**General.**
**Comment.** A clear limitation is an assumption (lines 95-96) that the coarser-resolution wider-area model is accurate at all its grid points…. However, the assumption leaves no scope for adjusting values at the coarser-model grid points. Thus the limited accuracy of the coarser model is "built in" and the method is strictly interpolation, albeit allowing for statistical properties of finer-scale fluctuations (anomalies). It seems to me that this is reflected in the validation (section 2.4) that the comparisons with OSTIA and ARGO data show very similar bias and RMS error for the coarser and finer models.
**Response.** This limitation has been removed in the revised manuscript by adding a new sub-section 2.3 Effect of noise in the input data. The calculations when the parent model is noisy (i.e. not accurate at all of its grid points) show that SDD method gives much better approximation to the true field that 'strictly interpolating' methods such as bi-linear or bi-cubic

interpolation of the coarse mesh data. For example in case of 10% noise in coarse grid, the RMSE error between the fine grid data generated by the SDD method and the true field is the same 10%, by bi-linear it is 25%, and by bi-cubic it is 19%. Moreover, we present an example where the coarse mesh is eddy permitting and the fine mesh is eddy resolving, with a resolution doubled in each spatial dimension. In this situation, mesoscale eddies are embryonic in the coarse mesh and can be restored into the fine grid.

**Comment.** Probably a finer-resolution model (impractical – the point of the manuscript) would be more accurate and give different results at the coarser-model grid points.

**Response.** The assumption that a finer model would be more accurate is not always the case, at least when standard point-wise metrics are used like RMSE and bias. The following quote from (Crocker et al. 2020) explains the situation (emphasis added):

> One of the issues faced when assessing high-resolution models against lower-resolution models over the same domain is that often the coarser model appears to perform at least equivalently or better when using typical verification metrics such as root mean squared error (RMSE) or mean error, which is a measure of the bias. **Whereas a higher-resolution model has the ability and requirement to forecast greater variation, detail and extremes, a coarser model cannot resolve the detail and will, by its nature, produce smoother features with less variation resulting in smaller errors. This can lead to the situation that despite the higher-resolution model looking more realistic it may verify worse (e.g. Mass et al., 2002; Tonani et al., 2019).**

> This is particularly the case when assessing forecast models categorically. If the location of a feature in the model is incorrect, then two penalties will be accrued: one for not forecasting the feature where it should have been and one for forecasting the same feature where it did not occur (the double-penalty effect, e.g. Rossa et al., 2008). **This effect is more prevalent in higher-resolution models due to their ability to, at least, partially resolve smaller-scale features of interest.** If the lower-resolution model could not resolve the feature and therefore did not forecast it, that model would only be penalised once. Therefore, despite giving potentially better guidance, the higher-resolution model will verify worse.

> The manuscript was amended to include the clarifying text and references (Lines 322-342)

**Comment.** Another assumption is that the distribution of fluctuations (anomalies, at any one depth) is statistically uniform and isotropic horizontally (line 129). This is inherently a limitation on the area of the (sub-)region where interpolation for finer resolution is desired. It may imply avoidance of nearby coasts, other distinct topography or water-mass boundaries (for example), despite the optimisation of weighting coefficients allowing for coasts.

**Response.** The assumption is actually that the distribution of fluctuations is statistically uniform and isotropic horizontally only locally, within the radius of computations given by Eq (8), not over the whole area. Clarification is added in Line 137. Such assumption is not

unusual. Modern data assimilation schemes assume statistical uniformity/isotropicity in the horizontal. For example, "The NEMOVAR ocean data assimilation system as implemented in the ECMWF ocean analysis for System 4" section 4.6.2 "Length scales" reads: "The horizontal background-error correlations for X = T , S U and η U are assumed to be isotropic poleward of a given latitude $\varphi_L$ , with an identical length-scale $L_\lambda = L_\varphi = L$ used for all variables and at all depths". ([https://www.ecmwf.int/en/elibrary/11174-nemovar-ocean-data-assimilation-system-implemented-ecmwf-ocean-analysis-system-4](https://www.ecmwf.int/en/elibrary/11174-nemovar-ocean-data-assimilation-system-implemented-ecmwf-ocean-analysis-system-4)) Some data assimilation schemes allow for non-homogeneity in the length scale, but *local* homogeneity is still required. This means that when computing the correlation matrix, homogeneity is assumed within the computation radius of every node.

**Comment.** Abstract.  It is important that the abstract is clear and easily understood.  Please clarify:

Line 14.  What is the "double penalty" effect?

**Response.** The double penalty effect is described in the literature as follows. If the location of a feature in the model is incorrect, then two penalties will be accrued: one for not forecasting the feature where it should have been and one for forecasting the same feature where it did not occur (the double-penalty effect, e.g. Rossa et al., 2008). Double penalty phenomenon is more evident in the high resolution models (Crocker, R., Maksymczuk, J., Mittermaier, M., Tonani, M., and Pequignet, C.: An approach to the verification of high-resolution ocean models using spatial methods, Ocean Sci., 16, 831–845, https://doi.org/10.5194/os-16-831-2020, 2020 ). The SDD method honours the data on the parent coarse grid and hence the spatial structure is anchored onto the coarse grid, therefore there is no additional spatial shift and no additional double penalty effect compared to the parent model. Clarification is given in lines 322-342 of the revised MS.

**Comment** Lines 20-21.  "areas smaller than the Rossby radius, where distributions of ocean variables are more coherent".  If the point about "more coherent" is necessary then what is more coherent with what?  Maybe small structures have internal coherence but their occurrence and scales are more likely to be stochastic, not coherent.

**Response.** Ocean fields are more coherent within 1-2 Rossby radii than between more distant points, so that mesoscale eddies of that size are sometimes called oceanic coherent structures, see e.g.(  G.I. Barenblatt et al (eds), 1992. Coherent structures and self-organisation of ocean currents. M.Nauka, 198pp. In Russian:  Г.И.Баренблатт и др. (ред). 1992. Когерентные структуры и самоорганизация океанических движений : М. : Наука, 198 с.  ISBN 5020008079;  F. J. Beron-Vera, M. J. Olascoaga, and G. J. Goni,2008. Oceanic mesoscale eddies as revealed by Lagrangian coherent structures. Geophysical Research Letters, Vol. 35, L12603, doi:10.1029/2008GL033957;
P.F.J. Lermusiaux and F. Lekien, 2020. Dynamics and Lagrangian Coherent Structures in the Ocean and their Uncertainties,
[http://web.mit.edu/pierrel/www/talk/pfjl_lekien_final_oberwolfach05.pdf](http://web.mit.edu/pierrel/www/talk/pfjl_lekien_final_oberwolfach05.pdf)) and the references in the MS . Clarification is added to the text ( LINE 21, 490).

**Comment.** Line 23.  1/24th degree from 1/12th degree is only a factor of 2 and begs the question of how much refinement the method works for.

**Response.** An increase of resolution by a factor of 2 (in each horizontal direction) increases the computational cost be a factor of 10 or more. The number of nodes is quadrupled (2x2) and the time step should be made 2 times smaller to comply with the Courant–Friedrichs–Lewy stability condition, which give the increase of number of computations by 2x2x2=8. The computation would require a larger number of computing cores and the overhead adds 20%-30% or more due to non-linearity of scaling. The cost of a relatively small HPC cluster is about £100K, so the purchase of 10 times larger computer can be a game-stopper. The SDD method adds a small number of calculations which can be performed even on a laptop computer. The efficiency of the SDD method is discussed in Section 2.2 of the revised MS.

**Comment.** Line 25. ". . cost function which represents the error between the model and true solution." In practical use the true solution is not available.

**Response.** "True solution" or "true state" is standard parlance in data assimilation when calculating a cost function. In many formulations, variables for the true solution are included, even if that true solution is never known (see for example R. N. Bannister, A review of forecast error covariance statistics in atmospheric variational data assimilation. I: Characteristics and measurements of forecast error covariances, 2008. Quarterly Journal of the Royal Meteorological Society, https://doi.org/10.1002/qj.339) . Clarification is given in Line 180.

**Comment.** Lines 167-168. "The correlation matrix is calculated . . . for each grid node on the fine mesh." This is possible where the true field is known (as here) but not in practical application unless there are data with resolution as good as on the fine mesh. Such data cannot come from the coarser model.

**Response.** The correlation matrix can be computed in practice using one of several methods (e.g. Hollingsworth and Lonnberg, 1986) which do not require knowing the true field. For instance, in H-L method, the true state is removed and only the errors remain by subtracting the background values from the observations. In this case, the only requirement is that the data is unbiased, the true state is not needed. In our paper eq. (7) is presented as a parametrised approximation by Fu et al. The text is additionally clarified below Figure 1.

**Comment.** Line 170. Surely the "final stochastic downscaling is carried out using" Equation (1) with the now-known pi. Eq. (7) was used earlier to calculate the correlation matrix.

**Response.** The text is corrected as advised (reference to eq(7) is replaced with Eq(1)).

**Comment.** Lines 245-247. Regarding the comment on lines 167-168, actual data for Eq (5) only exists at nodes of the coarser grid. Do the other 75% of points on the finer grid invoke the assumption that deviations $f'$ are statistically uniform and isotropic in the horizontal plane? Please clarify.

**Response.** This is correct. In common with the theory of 2D turbulence ( eg Rhines, P.B. 1975. Waves and turbulence on a beta-plane. J. Fluid Mech. 69, 417–443), the SDD model requires that all deviations are locally (within the search radius) isotropic and homogeneous. Additional clarification is added in Lines 136-137

**Comment.** Line 256. "previously considered" meaning nearest adjacent (node) already solved for?

**Response.** This is correct. We have amended the Ms to incorporate this clarification (line 305).

**Comment.** Lines 324-325. I think that one cannot argue from the accuracy of the idealised experiment in view of the question about data at nodes on the fine mesh (lines 167-168 comment). In the Red Sea example the finer-resolution model has accuracy very close to the coarser-resolution model and may well have more small-scale features (as figure 9 – yet to come – suggests) but it is not yet clear that "it also improves the accuracy of simulation."

**Response.** We meant that the SDD model has the ability to forecast greater granularity, variation, and extremes with respect to simpler interpolation schemes shown (bilinear, bicubic and spline). We have amended the Ms to clarify this point. (Line 250-253).

**Comment.** Line 373. What is the basis for "underestimates"?

**Response.** We meant that the coarse model shows lower values of gradients. The text is amended as requested.(LINES 431-432)

**Comment.** Line 381. "vorticity" should be "enstrophy"?

**Response.** Yes. The text has been amended (LINE 440).

**Comment.** Lines 416-417. Same comment as on lines 324-325.

**Response.** Please see our response to comment for lines 324-325.

**Comment.** Line 429. Repetition: "optimal . . optimised"

**Response.** Thanks. The text has been amended to avoid repetition (Line 489).

**Comment.** Line 430. "short range, comparable with the resolution of the parent model"; is this a limitation on the refinement from coarser to finer?

**Response.** SDD has been designed to improve on the results from Eddy permitting models. From this point of view, it is not a limitation of the model but rather its desired area of applicability.

**Comment.** Lines 475-480. I think the origin of "greater granularity" in the finer model should be further discussed. Conceivably the statistics are related to those determining the

correlation matrix etc. but is there any deterministic element in the small-scale (c.f. lines 521-522), or (more likely) in the seasonal variation of their statistics (line 479)?

**Response.** Yes, there is seasonally (monthly) variation of the statistics in the norm that is used to compute the innovations. The text is amended to emphasize this (Lines 541-543)

**Comment.** Lines 489-495.  The sign of vorticity can be biased (e.g. in coastal eddies) but does not show in enstrophy.  Has SMORS a basis for showing such bias? How does any such bias in its output compare with the best available evidence?  More enstrophy is likely in the finer-resolution model but does its increase take it significantly closer to the "truth" – is there evidence to test that?  Certainly the finer resolution in figure 14 presents a more convincing picture but it appears to add little except interpolation; all the features are embryonic in the coarse-resolution figure.

**Response.** We did not notice any bias in the sign of enstrophy in our calculations. It is correct that more enstrophy is likely in the finer-resolution model. More enstrophy is closer to the truth where the true filed is known as it was shown in the idealised experiment in Sections 2.2 and 2.3. In section 2.2 and 2.3 it is shown that the SDD method is significantly more efficient in recreating smaller scale features that common interpolation methods such as bi-linear or bi-cubic. It is correct that the SDD method is designed mainly to improve the eddy-permitting models where the smaller scale structures such as mesoscale eddies are only embryonic.

We thank Reviewer 2 for helpful comments.

**Responses to community comments on the manuscript 'High resolution stochastic downscaling method for ocean forecasting models and its application to the Red Sea dynamics'**

**Comment.** The mentioned "double penalty" effect should be expected in downscaling due to the fact that the coarse model has first of all a coarse bathymetry compared to the higher bathymetry of the downscaled model. Therefore, the spatial displace of hydrodynamical features should be appeared especially in areas with not smooth bathymetry.

**Response**. It is an interesting thought.  We have found that more detailed bathymetry and the coastline result mainly in the differences in the area integrated vorticity. Clarification is given in lines 420-423 of the revised MS.

In our calculations the double penalty was more of traditional nature as explained in Crocker et al. 2020: 'If the location of a feature in the model is incorrect, then two penalties will be accrued: one for not forecasting the feature where it should have been and one for forecasting the same feature where it did not occur (the double-penalty effect, e.g. Rossa et al., 2008). This effect is more prevalent in higher-resolution models due to their ability to, at least, partially resolve smaller-scale features of interest.' Clarification is added in lines 322-342 of the revised MS.

**Comment.** I propose to the authors to look the recent Red Sea paper here below and add the relevant citation: Hoteit, I., Abualnaja, Y., Afzal, S., Ait-El-Fquih, B., Akylas, T., Antony, C., Dawson, C., et al.  (2020). Towards an End-to-End Analysis and Prediction System for

Weather, Climate, and Marine Applications in the Red Sea. Bulletin of the American Meteorological Society, 1-61. https://doi.org/10.1175/bams-d-19-0005.

**Response.**  We are thankful for bringing our attention to the paper by Hoteit et al (2020). It is an interesting and comprehensive paper. It covers a wide range of topics many of which are outside the scope of our manuscript. We plan to work on some of the issues covered in this paper and add relevant discussions and citations in our future publications.

We thank Dr Zodiatis 2 for helpful comments

---

## Referee Report (RR1)

**Second Review** of "High resolution stochastic downscaling method for ocean forecasting models and its application to the Red Sea dynamics" by Georgy I. Shapiro, Jose M. Gonzalez-Ondina, Vladimir N. Belokopytov

**Overall comments**

The authors have given good responses to most of my previous comments. This review repeats the previous comments which I do not think have been adequately addressed and explains why I am concerned about them.

I have made three additional comments to those from my earlier review. I apologise that I did not make these comments in the first round but these points have only crystallised in my mind following more study of the paper and authors' responses.

I still think this paper is interesting and should be suitable for publication after significant revision provided the authors are willing to re-consider some of their claims.

**Previous comments**

1.  As I read it, the main idea in this paper is that optimal interpolation (OI) is the best method for interpolating a field $f$ of values specified on a set of regular grid points $x$, to another set of points. This may seem obvious from the name "optimal interpolation", but OI is not usually thought of in this context. It's usually thought of as a method for combining a number of irregularly spaced noisy observations with a climatology or model background to produce an optimal analysis. Figure 2 of the paper shows that, for some regularly gridded fields, OI provides much better interpolation results than standard methods like bi-linear or bi-cubic interpolation. If this is correct, which I think it is, it could be important for several reasons. First, in the data assimilation context, interpolating the model to the observations accurately is widely acknowledged to be a key step. So that doing that more accurately should improve the results. Second, as shown in figure 14, maps of fields with a lot of fine scale structure, such as the vorticity, may be rendered with greater fidelity using OI for interpolation rather than other interpolation methods.

    The authors did not respond to this (first) paragraph of my review which contained quite a number of relevant assertions and suggestions. My main point was that the proposed technique is simply an improved method for interpolation of fields based on the theory of optimal analysis. As the authors illustrate, this method can be successful even when the statistics of the correlation functions used are not very accurate (the statistics were very anisotropic and assumed to be isotropic). Optimal interpolation used in such a context is usually termed objective analysis. The method proposed is clearly not a dynamical downscaling methodology. Although, as the authors explain, optimal analysis can be considered to be founded on concepts derived from homogeneous turbulence and related to stochastic methods, describing the method as a stochastic downscaling will give most readers the impression that the method is based on multiple realisations with higher resolution, which is not really the case.

2.  Title: I wonder whether the words Optimal interpolation (or objective analysis?) should be in the title. For example: "Extraction of near grid-scale dynamical information from model fields using optimal interpolation." I'm a bit concerned that the main point of the paper is not evident from the title.

The authors did not respond to this comment either. My earlier comment implies that "objective analysis methods" would be a better wording than "optimal interpolation"

3. The literature on methods for post-processing of model outputs using Kalman filters should probably be discussed in the introduction. I think the main idea being pursued in this paper is somewhat different from the main ideas in that literature but the techniques are clearly related.

   My point here is that there is a literature on post-processing that is different from that on data assimilation. The authors could explore that by simply googling "Kalman filter post processing".

4. One might ask whether the method proposed is a post-processing of model output or a statistical model in its own right. It is described both as a Statistical Model (in SMORS) and a Stochastic Deterministic Downscaling (SDD) method. Personally I would view it as a post-processing method but do not feel strongly about this semantic issue.

   Authors' response: We prefer to term the SDD method as part of the model based on how it is implemented in the code. This is shown in the flowchart in Fig.4.

   From my previous comments it should be clear that I feel more strongly now that the method should be viewed as a post-processing step.

5. Figure 6: The lack of a double penalty …

   Your response to this point was very helpful. Thank you.

6. Lines 296-298: It seems strange to use nearest neighbour values in the ARGO inter-comparison. With the OI method one can do much better interpolations! Some readers may be concerned that the nearest neighbour method could somehow account for the lack of a double penalty (see previous paragraph).

   I'm quite concerned about this point. For many observations the nearest neighbour is further away on the coarse resolution grid. So the coarse grid value can be expected to be less accurate than the fine grid value. I think this gives the fine resolution grid model a significant advantage over the coarse grid one and could well be disguising a resolution penalty.

**New comments**

7. Lines 18-19 of the abstract state: "Then the method is applied to create an operational eddy-resolving Stochastic Model of the Red Sea (SMORS) with the parent model being the eddy-permitting Mercator Global Ocean Analysis and Forecast System."

   This claim that an eddy-permitting model is transformed into an eddy-resolving model is a significant exaggeration in my view.

8. I have tried to work out whether the proposed method truly provides some down-scaling. The following argument suggests that it does and that one would expect some form of down-scaling penalty to attach to it. It seems to me that the Fourier spectrum for the fine grid fields will be very close to that of the coarse grid fields down to the Nyquist wavenumber of the coarse grid. The fine grid will then have (probably small) non-zero amplitudes in the Fourier spectrum down to its Nyquist wavenumber. These intermediate Fourier amplitudes will be guided by the form of the correlation function. This additional power seems to me to be a modest form of down-scaling that could be based on the estimates of the statistics of the ocean fields.

9. The enstrophy field clearly has larger values on the fine than the coarse grid. I am not sure whether this relates to the additional power in the Fourier spectrum or the fact that the derivatives for the fine grid are calculated using smaller grid spacing than on the coarse grid.

---

## Author Response (AR2)

Responses to Reviewer 2/editor comments

*Dear Authors*
*Thank-you again for your revisions. I now have the referee 1 second set of comments, copied at the end of this in case you have not seen them. I am asking for "minor revision" in which you should please address my "Editor comments" and those of referee 1. In view of other correspondence between myself and the referee as well as with yourselves, the "Editor comments" raise a few points which should be brought into the open review process (on eventual publication all comments in the system are made public).*
*Yours sincerely, John Huthnance (Editor)*

*Editor comments*
**Comment.**

*I believe that we all (Referee 1, yourselves and myself) could agree on "optimal interpolation" (both words together) as a descriptor. Or maybe "objective analysis"? "Stochastic" and "deterministic" next to each other seems self-contradictory. "Stochastic" implies random (albeit obeying a distribution) whereas "deterministic" implies no randomness.*

**Response.** We were thinking of using 'objective analysis' instead of 'optimal interpolation', however , there are some pitfalls there.  We agree that the term 'optimal interpolation' may be confusing without a proper explanation as it is of a very different nature than the usual deterministic interpolation methods (linear, polynomial, spline, inverse distance etc) where the weighting coefficients are determined by the location of points, not by the data themselves. . In contrast the OI method calculates the weights based on statistical properties of the data. As to the term 'objective analysis', it  has already been occupied in the original publication by Cressman (1959) for his deterministic interpolation method. Therefore we decided to follow a convention from literature and use the term 'optimal interpolation' even though it is not strictly interpolation. It is known that OI method is a kind of minimum variance estimator that is algorithmically similar to Kalman filtering.

We agree that "Stochastic" and "deterministic" are antonyms, and we replaced the term Stochastic Deterministic by Stochastic-Deterministic in the MS. This format (using an 'en-dash') would be similar to the convention used in general physics e.g. 'wave-particle duality of light'. **Further clarification is given in the revised MS lines 57-63.**

**Comment.**  *"Double penalty as an issue" mainly applies to forecasts rather than analyses. Lines 337-344 nicely describe the "double penalty". However, this raises the question of the aim of your work; (i) interpolation or (ii) enhanced presence of small-scale features? For (i) you want to avoid "double penalty" if faithfulness to the parent model is important. For (ii) if you want small-scale but probably stochastic features with realistic frequency (spatial and temporal) and intensity, however obtained, then RMSE with liability to double jeopardy is an inappropriate measure.*

**Response.** The aim of the paper is to present an alternative downscaling method, i.e. better reveal small scale features of the fields by using stochastic properties of the data rather than use a 'brute force' method, i.e. run the same deterministic model but with more densely placed grid nodes. The reduction of the double penalty is just a nice side effect.

**Comment.** *This all relates to (a) clarifying the aims of your work and (b) using corresponding terminology. The present line 93 is relevant but aims should not be "buried" in a section headed "The algorithm". [It seems to me that your method may do best in going from eddy-permitting resolution where the desired features are "already" there embryonically and guided by assimilation (e.g. as in CMEMS) to somewhat finer resolution so that the embryonic features can be properly represented. I don't think your process has a basis for generating (ii) if the realistic frequency (spatial and temporal) and intensity of probably stochastic smaller-scale features are greater than what is already embryonic in the coarser-resolution product.]*

**Response.** Thank you for this comment which gives a very good and concise formulation on the applicability of the SDD method. **Assuming you do not mind, we copy-paste your words at the beginning of the Intro section,** to avoid them being buried in the Algorithm section.

Referee 1
*Second Review of "High resolution stochastic downscaling method for ocean forecasting models and its application to the Red Sea dynamics" by Georgy I. Shapiro, Jose M. Gonzalez-Ondina, Vladimir N. Belokopytov*

*Overall comments*
*The authors have given good responses to most of my previous comments. This review repeats the previous comments which I do not think have been adequately addressed and explains why I am concerned about them.*
*I have made three additional comments to those from my earlier review. I apologise that I did not make these comments in the first round but these points have only crystallised in my mind following more study of the paper and authors' responses.*
*I still think this paper is interesting and should be suitable for publication after significant revision provided the authors are willing to re-consider some of their claims.*

Responses to additional comments by reviewer 1

Previous comments.

**Comment.** *As I read it, the main idea in this paper is that optimal interpolation (OI) is the best method for interpolating a field f of values specified on a set of regular grid points x, to another set of points. This may seem obvious from the name "optimal interpolation", but OI*

*is not usually thought of in this context. It's usually thought of as a method for combining a number of irregularly spaced noisy observations with a climatology or model background to produce an optimal analysis. Figure 2 of the paper shows that, for some regularly gridded fields, OI provides much better interpolation results than standard methods like bi-linear or bi-cubic interpolation. If this is correct, which I think it is, it could be important for several reasons. First, in the data assimilation context, interpolating the model to the observations accurately is widely acknowledged to be a key step. So that doing that more accurately should improve the results. Second, as shown in figure 14, maps of fields with a lot of fine scale structure, such as the vorticity, may be rendered with greater fidelity using OI for interpolation rather than other interpolation methods.*

*The authors did not respond to this (first) paragraph of my review which contained quite a number of relevant assertions and suggestions. My main point was that the proposed technique is simply an improved method for interpolation of fields based on the theory of optimal analysis. As the authors illustrate, this method can be successful even when the statistics of the correlation functions used are not very accurate (the statistics were very anisotropic and assumed to be isotropic). Optimal interpolation used in such a context is usually termed objective analysis. The method proposed is clearly not a dynamical downscaling methodology. Although, as the authors explain, optimal analysis can be considered to be founded on concepts derived from homogeneous turbulence and related to stochastic methods, describing the method as a stochastic downscaling will give most readers the impression that the method is based on multiple realisations with higher resolution, which is not really the case.*

**Response.** The aim of the paper is to present an alternative approach to the commonly used 'brute force' method which uses the same ocean model but with more densely placed computational nodes in order to resolve small-scale features. The 'brute force' method is very expensive computationally at high (fine) resolution, while the SDD is computationally efficient. Therefore the main idea of the paper is to use a data assimilation method (we use the OI as an example, however could have used the Kalman filter or anything else) to create a finer -resolution field from a coarser -resolution parent model. As we explained earlier, despite its name, Optimal Interpolation is not a deterministic interpolation method like linear, inverse distance etc but minimum variance estimator that is algorithmically similar to Kalman filtering used in data assimilation. The SDD is a form of data assimilation with many similar features to the methods used in operational models. We have re-worded the introduction to make this clearer.

We have not found any evidence that statistics were anisotropic so we cannot comment on this statement. The statistics are assumed to be locally isotropic in line with many data assimilation methods.

The lack or reduction of double penalty error is an additional benefit of the SDD but it was not the primary purpose of the SDD development. The SDD method employs statistical properties of the data and hence we agree it is not purely deterministic, but it does not mean it is not dynamical. Quantum physics is a good example where the dynamical properties (e.g. movement of electrons or even nuclear explosion) are described statistically using the wave- (aka psi-) function which is a complex-valued probability amplitude.

We understand that the use of the original terminology (Optimal Interpolation) could cause confusion to the reader, and sometimes it is replaced with a 'objective analysis. However an issue has to be taken into account that the term 'objective analysis' was originally occupied ( eg Cressman 1959, Vasquez, 2003) a method which is a pure deterministic interpolation with weightings based on geometrical locations not on the statistical properties. It would have been more logical to call OI an 'objective analysis' and to call Cressman's 'objective analysis' a version of inverse distance interpolation. However we cannot change the history and use the terminology as it was introduced by its authors. We think that we could enhance confusion rather than remove it by changing the established terminology in our paper. **Clarification of this issue is given in Line 57-63 of the revised MS.**

**Comment.** *Title: I wonder whether the words Optimal interpolation (or objective analysis?) should be in the title. For example: "Extraction of near grid-scale dynamical information from model fields using optimal interpolation." I'm a bit concerned that the main point of the paper is not evident from the title.*
*The authors did not respond to this comment either. My earlier comment implies that "objective analysis methods" would be a better wording than "optimal interpolation"*

**Response.** We use the term 'Optimal interpolation' rather than more recent terms 'objective analysis' or 'Kalman filtering' in the body of the MS for historical reasons, namely we use the terms from the original literature of 1950s. We explained the method in detail in the paper trying to avoid some confusion in terminology (caused by the term interpolation) which was resolved 60 years ago. In modern literature, 'interpolation' generally refers to methods that allow to estimate magnitudes at new data points, based on the geometrical relationships between nearby points. In this case the weighting coefficients are computed based on points locations not on the data themselves. When these coefficients only depend on the point locations and, perhaps, a few parameters, the name 'interpolation' is appropriate.

The term Optimal interpolation is widely used in data assimilation schemes for ocean models ( see e.g. Pinardi, N., Allen, I., Demirov, E., De Mey, P., Korres, G., Lascaratos,A., Le Traon, P.Y., Maillard, C., Manzella, G., Tziavos, C., 2003. The Mediterranean ocean forecasting system: first phase of implementation(1998–2001). Annales Geophysicae 21, 3–20) whilst the term 'Interpolation' is not used in the title.

The use of the term 'interpolation' for a method in which the coefficients depend also on the data can be misleading. This is why we (and other researchers) use this term only in a fixed combination 'optimal interpolation'. The SDD calculates the weights based on the data which is similar to modern data assimilation schemes both from a fundamental point of view and from the mathematical and computational methods employed.

We believe that the title correctly represents the content of the paper. The method used is clearly described in the body and briefly in the abstract, and hence there is no need to include everything in the title.

The method we use is related to both dynamical and static downscaling, as it can be applied both to varying (dynamic) and static (stationary) ocean fields. The terms 'stochastic' or 'probabilistic' are not the antonyms to 'dynamic'. 'Stochastic' (based on data) can be

considered as opposite to 'deterministic' (based on equations) and both approaches are used in physics and mathematics. For example, in quantum mechanics the dynamic processes such as movement of electrons or even nuclear explosions are expressed in probabilistic terms. The main function describing the properties of elementary particles, so called wave function ( aka psi-function) is a complex-valued probability amplitude. The probabilistic approach using ensemble modelling is used in data assimilation in weather /ocean forecasting (e.g. Zanna et al, 2018, https://doi.org/10.1002/qj.3397). One of the major textbook on the subject has both 'random' (aka 'stochastic') and 'dynamic' in the title-L.Arnold, 1998, Random dynamical system, Springer, DOI: 10.1007/978-3-662-12878-7. **Clarification is given in lines 89-96**

**Comment.** The literature on methods for post-processing of model outputs using Kalman filters should probably be discussed in the introduction. I think the main idea being pursued in this paper is somewhat different from the main ideas in that literature but the techniques are clearly related.
My point here is that there is a literature on post-processing that is different from that on data assimilation. The authors could explore that by simply googling "Kalman filter post processing".

**Response.** The SDD method, in common to other data assimilation techniques, can be used both in the attached and detached modes. In the attached mode the downscaling is carried out on the same computer which solves the equations of ocean dynamics at the same time as the forecast advances. We use such mode for the operational Persian Gulf model which was developed jointly with the Met Office (not presented in this paper due to sensitive nature of the data it produces). Programmatically, in the attached mode the SDD is contained within the same executable module as all other elements of the model and is applied regularly as the model advances in time. On the other hand, in the detached mode, the SDD is applied after the forecast has been completed by the parent model. This mode was used in SMORS. In this case the SDD ( or any data assimilation) can be considered as post-processing. Some scientists consider data assimilation being not part of the model but part of the modelling system. The same applies to SDD. **We amended the text and added two references on the use of data assimilation (Kalman filter) as post-processing in the revised MS as advised.**

**Comment.** *One might ask whether the method proposed is a post-processing of model output or a statistical model in its own right. It is described both as a Statistical Model (in SMORS) and a Stochastic Deterministic Downscaling (SDD) method. Personally I would view it as a postprocessing method but do not feel strongly about this semantic issue.*
*Authors' response: We prefer to term the SDD method as part of the model based on how it is implemented in the code. This is shown in the flowchart in Fig.4.*
*From my previous comments it should be clear that I feel more strongly now that the method should be viewed as a post-processing step.*

**Response.** The SDD can be used both as part of the model run or at a post-processing stage. It seems that the boundaries between the core ocean model and pre-or post -processing are somewhat vague. Let us consider the NEMO model as an example. The vertical grid is generated by the model during the runtime using the subroutine domzgr.F which is supplied as an integral part of the model distribution code. However, philosophically, creation of the grid could be considered as pre-processing. The NEMO subsystem called XIOS (stands for XML-input-output server) can be run in two modes: attached and detached. In the attached mode the XIOS library is compiled into the same single executable file as the rest of NEMO, and each NEMO process acts also as a XIOS server. Hence it is considered as part of the model. In the detached mode every NEMO process runs as a XIOS client. Output is collected and collated by external, stand-alone XIOS server processors. In this case XIOS can be regarded as post-processing. Another example: is data assimilation an intrinsic part of a model or is it post-processing? The answer depends on what someone  means by the 'ocean model'. If it is just the set of primitive equations, then the answer is 'no'. If a model is wider collection of algorithms to simulate ocean processes, then probably 'yes'. Some modellers say that data assimilation is not part of the model but it is part of the overall modelling system. As SDD is a form of data assimilation, the same terminology can be applied to SDD. **We have modified the text clarifying that SDD can be used in both attached and detached (post-processing) mode in lines 89-96.**

**Comment.** *Lines 296-298: It seems strange to use nearest neighbour values in the ARGO intercomparison. With the OI method one can do much better interpolations! Some readers may be concerned that the nearest neighbour method could somehow account for the lack of a double penalty (see previous paragraph).*
*I'm quite concerned about this point. For many observations the nearest neighbour is further away on the coarse resolution grid. So the coarse grid value can be expected to be less accurate than the fine grid value. I think this gives the fine resolution grid model a significant advantage over the coarse grid one and could well be disguising a resolution penalty.*

**Response.** We agree with the statement '*So the coarse grid value can be expected to be less accurate than the fine grid value. I think this gives the fine resolution grid model a significant advantage over the coarse grid one and could well be disguising a resolution penalty.'* This is why we do the downscaling- to utilise the advantages of finer resolution grid.

We use the nearest neighbour method for compatibility reasons, as it is used for  validation of MyOcean / Copernicus Marine Environment Monitoring Service products, see e.g. Delrosso, D., Clementi,E., Grandi,A., Tonani,M., Oddo, P., Feruzza, G., Pinardi,N. 2016. Towards the Mediterranean forecasting system MyOcean v5: numerical experiments results and validation, 2016. INGV technical report, No 345, ISSN 2039-7941. The methodology presented in the report was developed by world leaders in the subject, and we follow it as good practice.  The full text of this report  is available from https://www.researchgate.net/publication/303973468_Towards_the_Mediterranean_Forecasting_System_MyOcean_V5_numerical_experiments_results_and_validation. **Clarification and an additional reference are given in lines 378-386 and 403-404.**

New comments.

**Comment.** *Lines 18-19 of the abstract state: "Then the method is applied to create an operational eddy resolving Stochastic Model of the Red Sea (SMORS) with the parent model being the eddy permitting Mercator Global Ocean Analysis and Forecast System."*
*This claim that an eddy-permitting model is transformed into an eddy-resolving model is a significant exaggeration in my view.*

**Response.** We have modified the text as advised. **The sentence is rephrased** avoiding the terms of 'eddy-resolving' and 'eddy-permitting' as follows : 'Then the method is applied to create an operational Stochastic Model of the Red Sea (SMORS) with the parent model being the Mercator Global Ocean Analysis and Forecast System at 1/12th degree resolution'.

**Comment.**. *I have tried to work out whether the proposed method truly provides some down-scaling. The following argument suggests that it does and that one would expect some form of down-scaling penalty to attach to it. It seems to me that the Fourier spectrum for the fine grid fields will be very close to that of the coarse grid fields down to the Nyquist wavenumber of the coarse grid. The fine grid will then have (probably small) non-zero amplitudes in the Fourier spectrum down to its Nyquist wavenumber. These intermediate Fourier amplitudes will be guided by the form of the correlation function. This additional power seems to me to be a modest form of down-scaling that could be based on the estimates of the statistics of the ocean fields.*

**Response.** We agree that the SDD method is prone to some form of down-scaling penalty. However, such penalty is significantly smaller than when using deterministic interpolation methods. We tried bi-linear and bi-cubic, however we think that a similar behaviour will be with other deterministic interpolators such as inverse distance, Sheppard's method, moving average etc. The actual values of downscaling errors, i. e. the difference between the downscaled or interpolated field and the true field are given in Figure 3 for the true field sampled with zero errors, and in Table 1 for a noisy true field. We attribute the better skill of the SDD method by the fact that it calculates the weighting coefficients based on the properties of surrounding data while the usual interpolation methods use weighting coefficients based only on the distance between the points no matter what the data are.

[Figure]

It is true that the Fourier spectrum of the field produced by SDD is close to the spectrum of the true field on the coarse grid up to the Nyquist wavelength of the coarse grid as can be seen on the figure below (also added to the revised MS). This figure shows the spectra of the following fields (i) true field on coarse grid, (ii) downscaled with SDD on fine grid, (iii) bi-cubic interpolated onto the same fine grid. The main peak on SDD spectrum is closer to the true peak than that produced by bi-linear interpolation. In the spectral region between the coarse and fine grids Nyquist wavenumbers, there is a parasitic peak which is an artefact caused by distortion of the fields downscaled by SDD as well as by bi-cubic interpolation from the coarse grid. However, this artefact is much smaller in the case of SDD which demonstrates its better skill of recovering the true field.

The idealised case where we know the true field gives us some confidence that the additional powers at high wavenumbers in the real world situation are mainly a representation of the true field, not artefacts.

We agree that the better representation of smaller scales by the SDD is due to resolving additional range of wavenumbers between the Nyquist values for coarse and fine grids.

**We have amended the text to reflect the above (lines 255-265 in the revised MS) and added a new figure (now Fig 4)**

**Comment.** *The enstrophy field clearly has larger values on the fine than the coarse grid. I am not sure whether this relates to the additional power in the Fourier spectrum or the fact that the derivatives for the fine grid are calculated using smaller grid spacing than on the coarse grid.*

**Response.** We think that the two effects are closely related. The additional range in the Fourier power spectrum is a reflection of smaller grid spacing and hence a potential of having sharper gradients. **We have clarified this is the text, lines 579-581.**

---

## Author Response (AR3)

Dear Editor,

We have implemented all suggested corrections. We think that the problem with faulty lines in the PDF was due to the comma sign was too close to the equation. We inserted a few spaces.

Regards

Georgy Shapiro